# How ornithopters can perch autonomously on a branch

Raphael Zufferey [1,2] ✉, Jesus Tormo-Barbero[1], Daniel Feliu-Talegón [1], Saeed Rafee Nekoo [1], José Ángel Acosta [1] & Anibal Ollero [1]

Flapping wings produce lift and thrust in bio-inspired aerial robots, leading to quiet, safe and efficient flight. However, to extend their application scope, these robots must perch and land, a feat widely demonstrated by birds. Despite recent progress, flapping-wing vehicles, or ornithopters, are to this day unable to stop their flight. In this paper, we present a process to autonomously land an ornithopter on a branch. This method describes the joint operation of a pitch-yaw-altitude flapping flight controller, an optical close-range correction system and a bistable claw appendage design that can grasp a branch within 25 milliseconds and re-open. We validate this method with a 700 g robot and demonstrate the first autonomous perching flight of a flapping-wing robot on a branch, a result replicated with a second robot. This work paves the way towards the application of flapping-wing robots for long-range missions, bird observation, manipulation, and outdoor flight.

Flight is energy-intensive, and no bird exists without some sort of perching system. In nature, the ability to land on a variety of surfaces is essential for most birds to hunt prey, watch reproductive sites, rest between movements in the landscape, or to monitor territories[1,2]. In flapping-wing robots, branch perching would also open a vast array of applications[3] for this class of robots. To start, they are ideal candidates to monitor wildlife, as their quiet and propeller-less operation has a lower impact on the environment[3]. From a branch, robots can observe and track animals both on the ground and in flight. Physical interaction with a tree could permit microscopic analysis of the branch's surface as well[4]. Sample return of a leaf can be envisioned, enabling biologists to study those systems with minimum collection effort. Energy recovery is an interesting possibility to extend the operation time of robots. Solar charging could enable flapping-wing robots to travel on longer missions[5,6]. Additionally, the capability to perch will enable many other applications, e.g. perching on pipes, power lines, and other structures for contact inspection[7]. Amongst all unmanned aircrafts, flapping-wing robots offer unrivaled safety operation making them suitable for interaction with humans, animals, plants and even industrial structures.

While the prospect of this technology is high, achieving localized perching from flapping-wing flight is challenging. The hard-to-model,

unsteady aerodynamics of the flapping-wing motion lead to less accurate control and therefore less accurate positioning. Small-scale birds and robots that are capable of hovering circumvent this issue[8,9], however they suffer from limited payload and increased manufacturing complexity. On the other side, large flapping-wing robots and birds are unable to hover due to unfavorable scaling, and therefore need forward velocity to maintain flight and be controllable. Consequently, landing on the branch requires a grasping method capable of stopping a forward-moving flapping robot. This is challenging due to the combined requirements of high-speed actuation, precise timing, and high impact resistance. Importantly, the oscillations in altitude, induced from the flapping-wing motion, should be compensated by the grasping appendage, which has to tolerate misalignment. Moreover, perching is more difficult for large (> 1 m wingspan) ornithopters. Indeed, appendage strength scales with the cross-sectional area $L^2$, but the kinetic energy of the flying vehicle scales up much more strongly with $L^3 v^2$[10]. This implies that $Lv^2 \sim 1$. Therefore, the bigger the robot, the slower the landing speed needs to be but $v$ has a lower limit in order to maintain flight. Additionally, once landed, the grasping mechanism needs to hold the robots' weight, as confirmed by[11]. The weight scales with $L^3$ but the gripping strength depends on the claw's force which scales with $L^2$, also limiting how large perchers can be.

[1]GRVC Robotics Lab., Departamento de Ingeniería de Sistemas y Automática Escuela Técnica Superior de Ingeniería, University of Seville, Seville, Spain. [2]Ecole Polytechnique Fédérale de Lausanne (EPFL), Lausanne, Switzerland. ✉e-mail: raph.zufferey@gmail.com

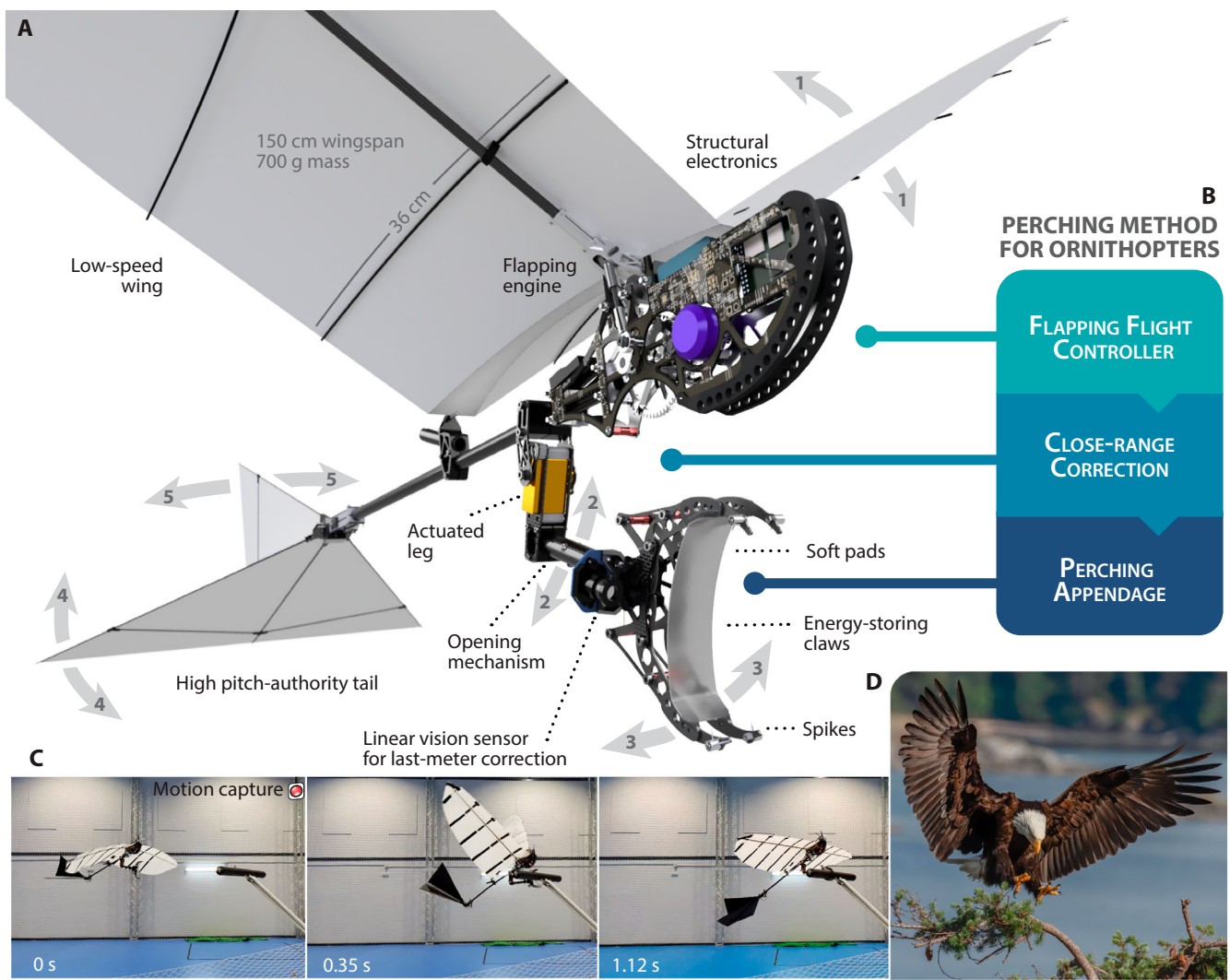

**Fig. 1 | Overview and demonstration of the perching method. A** Rendered view of the leg-claw system with the flapping-wing robot. The five degrees of freedom are displayed as well as the features necessary for perching flights. **B** The steps of the proposed perching method for ornithopters. **C** Perching sequence. **D** Eagle shortly before landing on a branch by D. Freeman, CC BY-SA 2.0.

Many solutions to perching unmanned aerial vehicles exist, and have been extensively reviewed[12]. One example, a fixed-wing prototype capable of perching to walls was proposed in[13]. Attachment to a surface is an important issue, addressed differently with systems such as micro-spines[14,15], spines[16], fiber-based adhesive[17], or nature-inspired mechanical grippers[18,19]. Multirotor UAVs are able to hang passively from branches of various diameters[20], or sit passively on branches[21]. Attachment under beams was also shown with quadcopters, based on a bistable clamping mechanism[22]. Researchers have also investigated how to perch robots to point locations. For example, multirotor UAVs, capable of hovering, were employed to perform localized perching with vision-based feature identification[23,24]. Roderick et al. recently presented a bird-inspired perching mechanism, installed under a multirotor[25]. This robot features impressive perching capabilities without needing accurate control to perch using a tendon-driven locking mechanism for passive perching. This robot possesses high graspability thanks to the multi-joint claw design. The designs mentioned above are mounted on a multirotor UAV, which are capable of hovering as well as carrying heavier payloads, unlike flapping-wing robots. In addition, ornithopters require an impact-resilient leg-claw system capable of stopping the fast-moving vehicle. Drone perching tasks are performed from above thus not requiring significant unbalance resistance. The maneuver can happen as slow as needed and the

position precision is superior to that of a forward flying robot. Very recently, Stewart et al. succeeded in perching fixed-wing robots from a catapult[26]. This research presented impressive grasping capabilities at up to 7.4 m/s, widening the field of operation of fixed-wing robots, but perching from free flight is yet to be demonstrated. Compared to fixed-wing robots, flapping-wing robots face additional constraints such as increased payload restrictions and oscillations that need to be addressed. Overall, physical interaction and specifically perching is generating strong interest in robotics, highlighted by excellent recent research applied to ornithopters[19].

To the best of our knowledge, the only flapping-wing robot capable of perching is the centimeter-scale Robobee[27], which weighs 100 mg. Perching at this scale differs widely, i.e. it was executed under a flat ceiling, leveraging electrostatic adhesion. To this day, no large flapping-wing robots have demonstrated perching on a branch. Yet the potential of this technology is vast as such robots are good candidates for aerial physical interaction and manipulation.

In this paper, we propose a novel autonomous perching method capable of landing and maintaining large flapping-wing robots on a branch. This three-phased method consists in the simultaneous operation, within a flying vehicle, of a flapping-flight controller, a close-range correction system and a passive perching appendage. The perching method is validated with a 1.5 wingspan − 700 g robot,

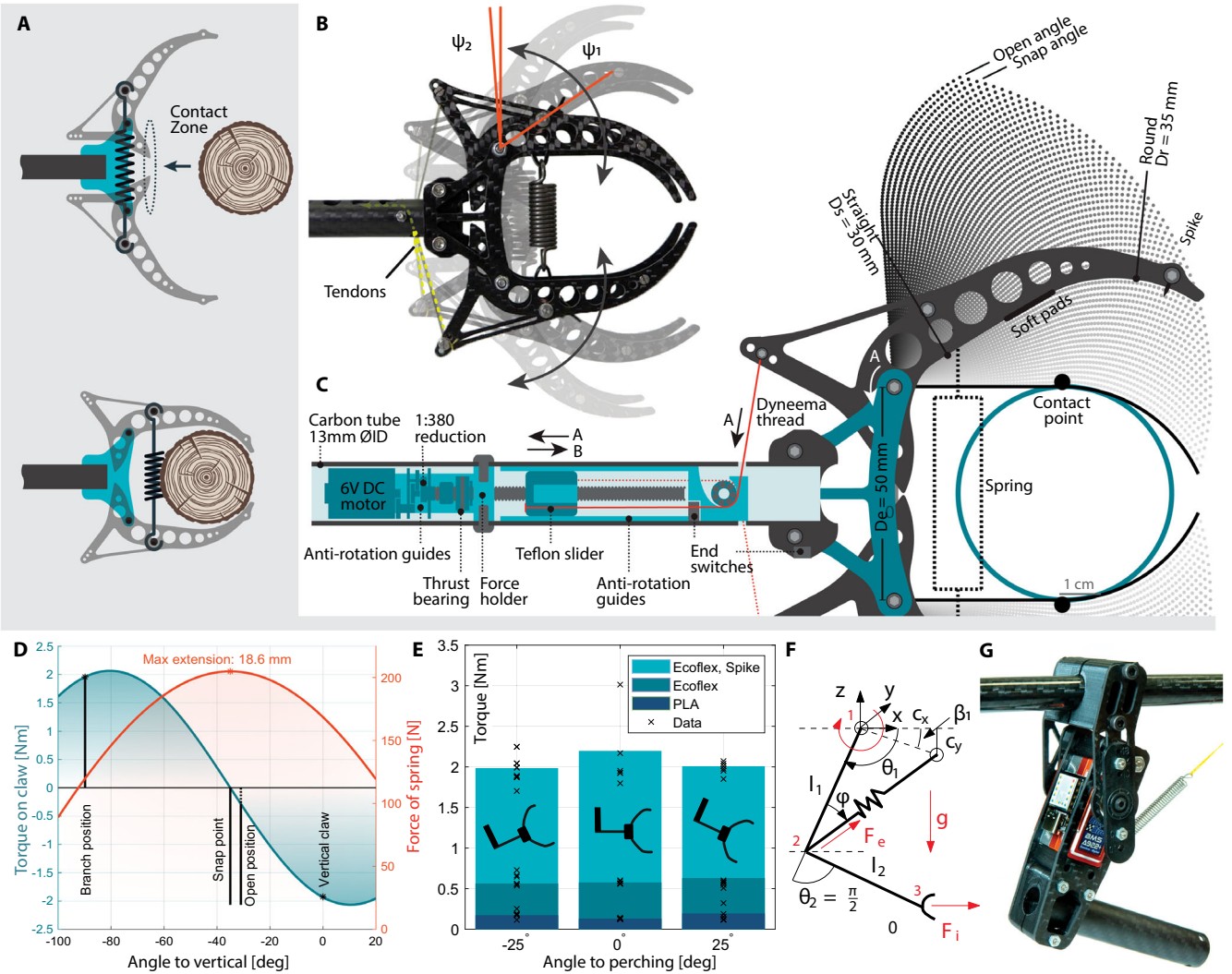

**Fig. 2 | Design of the leg-claw appendage. A** Bistable claw configuration, before and after contact. **B** Photo composition of the claw in open, closed, and in-between positions. Open angle is $\Psi_1$ and closed angle is $\Psi_2$. The visible spring absorbs the impact. **C** Tendon-pulling re-opening mechanism entirely integrated within the leg tube, see a rendered section view in Supplementary Fig. 12 and modeled geometry of the inner part of the claw shown at increasing angles between the open and closed/contact positions. **D** Projected torques and forces on the branch. **E** Torque measurement of the claw with three different contact materials (Spikes, Ecoflex silicone and PLA plastic). **F** Leg geometry parameter definition for the modeling of the leg. **G** Side-view of the optimized leg assembly.

depicted in Fig. 1A, resulting in the first indoor autonomous branch perching demonstration of a large flapping-wing robot. The flight-perch maneuver is performed autonomously with localization from a motion capture system, ending on a wooden branch at a 2 m altitude, see a time-lapse in Fig. 1C. The level of autonomy achieved in this study is without user input during the entirety of the experiment, fully on-board computation and navigation, but dependent on accurate localization data from a motion capture system. We also present the development process, consisting of four experimental steps, which leads to autonomous perching flights. Overall, this paper discusses the challenges, the robotic methods and an implementation that performs full flight and perching.

## Results

Flapping-wing robots are human's current best effort at replicating bird locomotion, and it is therefore logical to examine landing birds (see Fig. 1D) for insights about this complex maneuver. As birds approach a landing target, their body pitches up, effectively reducing the flight speed drastically. They then point their extended legs forward and unfurl their claws towards the perch based on visual perception and accurately position their body within the estimated time

to contact[28]. The landing occurs above the branch, unlike hanging bats. Birds rely on a set of claws, actuated by powerful muscles which can apply forces up to 2.5 times their body weight. Their claws can undergo ultra-fast grasping motion, on the order of 1 ms, which hints at stored potential energy in their tendons[29]. They secure the animal without energy expenditure and with high friction thanks to the combination of hard claws and soft toe pads. These extracted cues from biology have been directly employed in the robotic method presented here. Our proposed perching method consists of the simultaneous operation of three key components necessary for a successful maneuver, as shown in Fig. 1B.

First, the flight control system needs to handle stable flight with the appendage payload and all the on-board electronics required for flight and perch. The flight controller should bring the robot sufficiently close to the branch, within the range of the close-range correction system. This is possible with a pitch-yaw-altitude controller. A printed circuit board (PCB), fitted with a companion computer, should handle the autonomous flight and perching operation. Flight tests allow for tuning and validation of the controller.

Second, the claw is held by an actively controlled leg which corrects for the vertical position inaccuracy inherent to flapping-wing

flight as the robots approach the landing target. Feedback is enabled thanks to optical sensing of the branch at high speed, resulting in a close-range correction system.

Third, the claw appendage is mechanically triggered by the branch itself, and has to enable perching at up to 4 m/s. The mechanical appendage of the robot needs to be able to resist the impact, close quickly onto the branch, apply sufficient force to remain in place, and lastly tolerate misalignments.

The first implementation of the perching method resulted in the ornithopter P-Flap for Perching Flapping-Wing Robot, see the specifications in Supplementary Table 1. This design is initially based on the pre-existing E-Flap robot[30]. This flapping-wing vehicle features 100% payload capacity and offers stable flight velocities as low as 3 m/s, both possible thanks to the lightweight construction and low wing loading of 16 N/m². This class of robots is large, with a 150 cm wingspan and a 500 g empty weight. This size represents a good trade-off between manufacturing and testing and scaling constraints, i.e. the robots utilizes commercial components, widely-employed prototyping methods yet is small enough to fly within a motion capture lab space.

### Bistable claws as a perching mechanism

The landing of a large-scale flapping robot on a branch requires a grasping appendage. This extremity mechanism should be capable of rapid actuation at the adequate moment and exertion of a sufficient force to counteract the rotational inertia upon landing and imbalance position thereafter. A direct DC drive system of the claw would have a negative impact both in terms of additional mass, resulting in a higher flight cost and power draw during perch. Indeed, the robot should remain on the structure without energy expenditure to enable long-duration manipulation and observation tasks which would be unfeasible under constant power draw.

We propose a new claw-leg system based on a double row of carbon-fiber plates that rapidly close onto the branch upon contact. The provision of stored energy permits both high speed and high force. Concurrently, this claw design offers low mass, impact-release, and self-locking, thanks to the bistable topology, shown in Fig. 2A. The claw is in the open (top) position before impact. This position is stable with a negative $\psi_2 = -4°$ angle, i.e. the spring origins are located behind the centers of rotation. The $\psi_2$ angle is fixed by the carbon-fiber design resting against the tube holder and end-position switch. In the open position, two protrusions in the central part of the claw act as triggers for actuation. As the branch gets in contact with the claws, it can either directly impact the protrusions or be off-centered and impact the extremity of the claws. In this second case, the claw will then slide along the angled claw until it contacts with the central protrusion, also triggering the closing of the claw. As the claws pivot inwards onto the branch, the spring passes frontwards of the center of rotation (see Fig. 2B) accelerating until the claws contact the branch. The spring here plays a simultaneous function as a shock absorber, reducing the load on the carbon-fiber side plates. Once fully closed, the claw continuously applies a force of 56.8 N on the branch, at the contact point.

### Claw geometry

The geometry of the claw is modeled to offer maximum clamping force at the branch level while also having a wide opening angle to capture the branch and sufficiently low trigger force to ensure reliable closing on impact. The branch-claw contact line is composed of a straight $D_s$ segment followed by a rounded section with a $D_r$ radius, presented in Fig. 2C. These values are optimized to ensure a contact point close to the center of rotation, increasing the applied force on the branch. The $R_c$ radius is slightly larger than the selected branch diameter. This guarantees contact of the spikes and accepts branches with larger diameters. The rotation points of the two claws are spaced by $D_e = 50$ mm, setting the minimum branch diameter with

spike contact. The origin of the spring in the claw is calculated so that the maximum torque on the claw is reached when closed, shown in Fig. 2D. The maximum permissible extension is reached at the snap point when the spring passes the claw rotation center. In the open position, the torque of the claw is small, below 0.2 Nm. This results in a required force to release of 11.4 N at the tip of the contact zone, well below the forces encountered on impact, even at low speeds.

### Claw locking

The torque that the claw can withstand before slipping is determined by the interface materials and contact between the carbon-fiber frame and wood. Figure 2E shows comparative experimental results of various interface methods. A combination of hard tip claws and soft toe pads is expected to give the highest friction[29]. The Ecoflex-covered toe pads are designed to cover the interior part of the claws in contact with the branch. The Ecoflex membrane additionally covers the spring. The metal spikes chosen in this work are found suitable to perch on branches and flat surfaces, as previously demonstrated[13,20]. Note that birds' claws are softer, and may perform better on a larger variety of branches shapes and surface textures[27]. See "Methods" for assembly details. As visible in Fig. 2C, the idealized branch location is tangent to the cylindrical axis of the spring, resulting in an overlap of the spring's diameter. This is the third contact point of the claw. However, in reality, the spring is forced leftwards. This has the benefit of continuously applying a horizontal force towards the spikes, improving grip through a pinching action. The video of high-speed claw closing is presented in Supplementary Movie 1. While the claw is designed around a 6 cm branch diameter operating point, it is tolerant to deviation from that value. In the 4–7 cm range, the locking capability is similar (within 10%). Further increasing the diameter degrades the claw force on the branch, remaining effective at 11 cm, confirmed analytically and in manual outdoor tests (Supplementary Figs. 15 and 16).

### Claw opening

While storing mechanical energy in the spring is a lightweight method to achieve a strong, fast grasp, it requires a re-opening mechanism capable of initially loading the claw and secondly releasing the grasp from the branch after perching. Whereas closing the claw needs to be a fast motion, re-opening can be reasonably slow. A high-reduction mechanism is therefore particularly well-suited to this task. We propose a perching claw that fully integrates a re-opening subsystem within the leg shown in a section view in Fig. 2C. This drive system operates in a pull leadscrew configuration and can pull up to 200 N of force (see tension experiment Supplementary Fig. 1). Opening of the claws occurs within 20 s, with the motor consuming an average of 3 W. Once fully open ($\psi_2$ position), an end switch stops the motor. The microcontroller then winds the carriage back down, by rotating the motor in the opposite direction. The tendons become loose, but the claws stay in an open position thanks to the bistable disposition (Supplementary Movie 2).

### Active leg

**Leg design and optimization.** The leg is actuated to perform three functions: misalignment compensation to compensate for the flight inaccuracies and disturbances shortly before landing; enabling changes in the pose of the robot just after the perching maneuver to maintain an erected pose on the branch and not fall; maintaining the equilibrium once the system is perched and the system performs manipulation tasks[31]. Considering these functionalities, the leg mechanism needs to meet hard constraints: the size and weight of the mechanism must meet the requirements of the flapping aerial robots; adequate impact resistance to withstand the perching forces and absorb part of the energy; actuation of the mechanism should be as precise, powerful, and fast as these functionalities require. The central body of the leg mechanism consists of two parallel sets of carbon-fiber plates. In-between the plates, a 34 g servo motor produces rotational

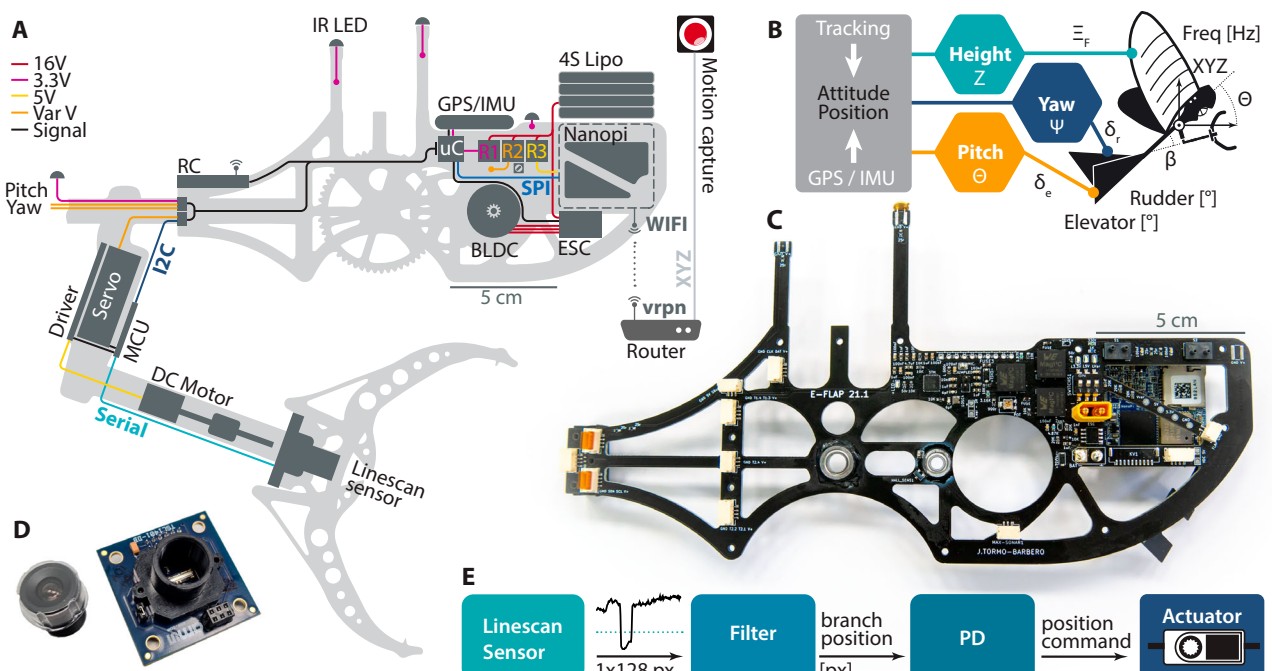

**Fig. 3 | Avionics and control of the P-Flap. A** Electronics layout of the perching robots, showing components and connections. This includes communication buses (Serial, I2C, SPI, VRPN), active electronics (microcontrollers MCU and uC, regulators R1, R2, R3, light-emitting devices IR LED) and stand-alone components (inertial-measurement unit GPS/IMU, brushless electric motor BLDC, electronic driver ESC and battery (4S Lipo)). **B** Triple control strategy that running on the companion computer. The control inputs are the position XYZ and attitude (only pitch angle Θ shown here). The outputs are the throttle $\Xi_F$, the rudder deflection $\delta_e$ and elevator deflections $\delta_r$. The leg angle is also regulated along $\beta$. **C** Structural PCB concept that composes half of the body. **D** Optical line-scan sensor TSL1401. **E** Last-meter leg control strategy.

motion between the body ornithopter and the leg-claw mechanism, see Fig. 2G. A diagonal spring between the leg tube and robot body stores energy upon impact, reducing the maximum stress that the mechanism suffers. The leg actuation mechanism is located below the robot body so that leg rotates between 0° (vertical downwards) and 90° (horizontal), close to the center of gravity of the system.

We developed a dynamic model of the system to calculate the parameters shown in Fig. 2F that minimize the damage upon impact during the perching maneuver, described in Supplementary Method 2 and 4. The dynamic model estimates the motion of the system and the stresses in the joints for a given impact force (Supplementary Fig. 4). A finite-element-based multi-body dynamics analysis validates the proposed model during the impact. In those simulations, the total mass of the system was considered 700 g and the speed of the ornithopter before the impact, varied in the range 2–4 m/s showing the accuracy of the proposed model. The combination of this dynamic model and an optimization process allows us to obtain the most appropriate parameters for the leg mechanism within a logical range. We define a cost function that considers the stresses in the servo, the angular momentum in the servo joint, which is related to the gear tooth stress, and the mass of the leg, displayed in Supplementary Fig. 3. The particle swarm optimization (PSO) is used to find the best set of variables (Supplementary Table 2) that satisfies the defined cost function (given in Supplementary Method 3).

**Leg control and branch detection.** The mechanical contact zone of the claw is 5 cm, which is lower than flight altitude accuracy, previously estimated to ±10–15 cm. As this is insufficient to reliably perch on the target, we propose a correction method. The leg of the robotic bird is articulated at the elbow level, permitting semi-vertical motion of the claws. As the robots approach the branch, the leg detects it and compensates for misalignment. The relative position of the branch is obtained from a line-scan sensor, shown in Fig. 3D. This class of sensors relies on a single-row pixel array which enables contrast-based detection[32]. Future outdoor perching maneuvers in cluttered environments will benefit from improved sensing methods, e.g. 3D vision systems[25], stereo CMOS-based or event-based cameras[33,34].

As shown in Fig. 3E, the line readout is filtered and the location of the peak is determined. The peak pixel offset is directly fed to a proportional-derivative (PD) controller to command the position of the servo-actuator, which moves the claw and the sensor on it, closing the loop. Therefore, the calculated offset of the branch relative to the claw is compensated by the mechanical motion of the elbow joint, which allows us to compensate for inaccuracies in the flight control. The gains of the controller are adequately tuned and three different experiments are carried out to verify the effectiveness of the approach; performing oscillations of high amplitude at 2 Hz shown in Supplementary Movie 3, oscillations of low amplitude at 4 Hz, and launching the ornithopter at a speed of 2.5 m/s with the launcher, see Supplementary Movie 4. The movement of the flapping-wing robots in flight can be described as a combination of these movements. The data obtained from the sensor provides reliable detection of a 6 cm-diameter branch at a distance of up to 190 cm, validated experimentally (see Supplementary Fig. 6).

## Robot avionics and control

Performing an indoor autonomous perching maneuver with an on-board active claw system requires tight integration of the electronics within the robots, with minimum connections, cables, and failure points. We leverage a full-body printed circuit board for this purpose, shown in Fig. 3C. The 1.6 mm-thick PCB doubles as a structural component, paired with the left carbon-fiber plate. Through this board, connectors are judiciously located to reduce cable length. Mechanical fabrication and assembly are vastly simplified as well as electrical troubleshooting. Control of the flying robots is split between a low-level custom controller (uC) and a high-level wireless companion

Fig. 4 | Experimental development process. Presentation of the process that leads to autonomous perching flights.

computer (Nanopi Neo Air), shown in Fig. 3A. The trajectory decision is calculated by the companion computer based upon the localization data transmitted by the motion capture system.

During the flight, three parallel control loops run in the custom autopilot at 120 Hz, implemented in C++. The three controlled states of the robot are pitch $\Theta$, yaw $\Psi$, and height $Z$, with their corresponding actuation elevator $\delta_e$, rudder $\delta_r$ and the flapping frequency $\Xi_F$, respectively (see Fig. 3B). An extra control loop for the leg-claw is activated during the approaching phase, close to the branch, that indirectly regulates the angle $\beta$ in Fig. 3B. Since the robot flight dynamic is highly nonlinear for this type of perching maneuver, the control strategy is designed ad-hoc to make it feasible with simple feedback loops. The steps are (1) Analysis of the flight envelope for different launching speeds through experimentation; (2) Selection of a suitable trajectory in which no control limitations appear, such as saturation or limited stability margins. Aggressive maneuvers or increases in height of more than 3 m in the 14-m longitudinal test-bed workspace are unsuitable trajectories that would force the flapping frequency and tail into saturation and miss the target. Therefore, the arrangement of the initial condition of the robot on the launcher, launching speed, angle, and desired Z height position should be reasonably chosen; (3) Tuning of the flight control loops around that specific trajectory with the height and perching velocity as performance criteria; (4) Tuning of the leg-claw approaching phase. It is stressed that steps (1) and (2) are critical because they ensure that the robot flies within its flight envelope with sufficient control authority for a PID-type controller and this maneuver. However, current and future work are underway to analyze more complex control architectures as e.g. those already implemented on a claw-less robot[35].

While the tailed ornithopters in this research are passively stable, it is challenging to reach accurate positioning during the perching maneuver at low speed. The difficulties are loss of control authority and large vertical oscillations. Indeed, flapping is needed to maintain a stable flight resulting in oscillations in the controlled height. As in most fixed-wing aircraft, the altitude dynamics are non-minimum phase, which inherently limits the achievable settling time. Coupling both makes the control of this maneuver a challenge. To show this behavior, the altitude dynamics result for a typical perching is shown in Supplementary Fig. 7, complemented by a sensitivity analysis (Supplementary Fig. 17).

### Launcher experiments
Achieving simultaneous and reliable operation of all three phases of the perching method is extremely challenging due to the high speed of the maneuver (typically less than 4 s) and the consequences of failure (damage to the robots). We therefore propose a 4-step development process that enables safe, iterative design, development and tuning of all the systems, see Fig. 4. First, flight is not considered and the forward velocity is given instead by a launcher, enabling investigation of the grasping appendage. Second, the autopilot controller is tested and tuned in flight without any leg or claw. Third, the leg and claws are added and the close-range correction system is validated with a soft branch. Fourth, full perching maneuvers can now be performed on the real branch.

A repeatable launching system was developed to measure the branch interaction characteristics, as it accelerates the robotic claw

or the full robot to a predefined velocity, shown in Fig. 5B. First, experiments with only the claw were performed, but loaded with equivalent bird mass. The claw separates from the launcher at its extremity and contacts with the branch after a short, 20 cm-distance air phase. The branch is fitted with a force sensor and perpendicular high-speed video was recorded. The claw concept and geometry were validated and optimized through successive perching tests, exemplified in Fig. 5A. We report on the impact resistance of the claw system by incrementally increasing its forward velocity. The experiments show a 100% perching success rate at up to 4 m/s with a fully flight-equipped robotic bird. No structural damage has been observed until this limit, which sits above the expected flight requirement of 3 m/s. Experiments show that the claw locks onto an object within 25 ms, slightly above the expected 20 ms. This is sufficiently fast for perching maneuvers. Indeed, impact tests (see Fig. 5C) show the time between the start of the impact and the start of the bounce to be 50 ms, see Supplementary Movie 5. Vertical misalignment, $z$, behavior is measured at 4 m/s speed, the highest expected flight velocity on impact. In all cases, forces remain below 150 N, see Fig. 5D, E

Leg pitch orientation, speed at the time of perching and yaw-deviation between the vector speed and the branch on perching success are three parameters that define the conditions required upon reaching the branch to successfully land. Their effect is experimentally investigated with the launcher system. The robot's pitch angle is fixed at 30° in all the experiments, comparable to the pitch angle achieved in flight. Figure 5F shows the yaw angle between velocity vector and branch effect on perching performance when the vehicle reaches the branch without being perpendicular to its axis. The results show that the system can tolerate a branch deviation about yaw in a range of ±10-20°, which also depends on the leg pitch orientation, where 90° is a horizontal leg. Therefore, in case of a small yaw angle deviation from the perpendicular direction, the claw can still adequately hold onto the branch, at 4 m/s perching speed. In Fig. 5G, the leg angle versus impact velocity is analyzed; note that in all cases the robot remains attached to the branch, but only in the "perched" state does it remain erect. Increasing impact velocities produce high angular momentum that overpasses the friction force and leads to forward fall, whereas lower impact velocities produce low angular momentum and high center of mass offset, leading to backward falls.

### Flight experiments
Landing onto a localized object demands accurate flight control. A 6 cm-diameter branch represents a distance 25-times smaller than the wingspan of the P-Flap, a precision difficult to match for a flying aircraft. The flight experiments were run in three sub-phases, shown in Fig. 6B. Initial tests (flight only) were performed without any branch, resulting in a safer environment for measuring flight performance, tuning the flight controller, and validating communication and electronics in-flight. Subsequent tests were performed with a soft branch, shown in Supplementary Movie 6. In this phase, a black foam cylinder held by detachable lateral velcro straps was used as a landing target (soft branch). This permitted observation and tuning of the active leg in flight whilst avoiding hard impacts with a rigid element. There, the claw repeatedly reached the soft branch. We performed the final experiments with a real wooden branch (branch / full perching), with typical real-time flights shown in Supplementary Movie 7 and a slow-motion view in Supplementary Movie 8. The 80-cm-black-painted branch is held at 2 m height at its center by a 45° aluminum profile that reaches out through the safety net.

The forward flight velocity is a key parameter for perching. It should be kept low to minimize impact forces and increase reaction time before impact. To achieve this, the pitch angle of the robot needs to be high. As the pitch increases, so should the flapping-wing frequency to maintain altitude. The powertrain, therefore, limits the

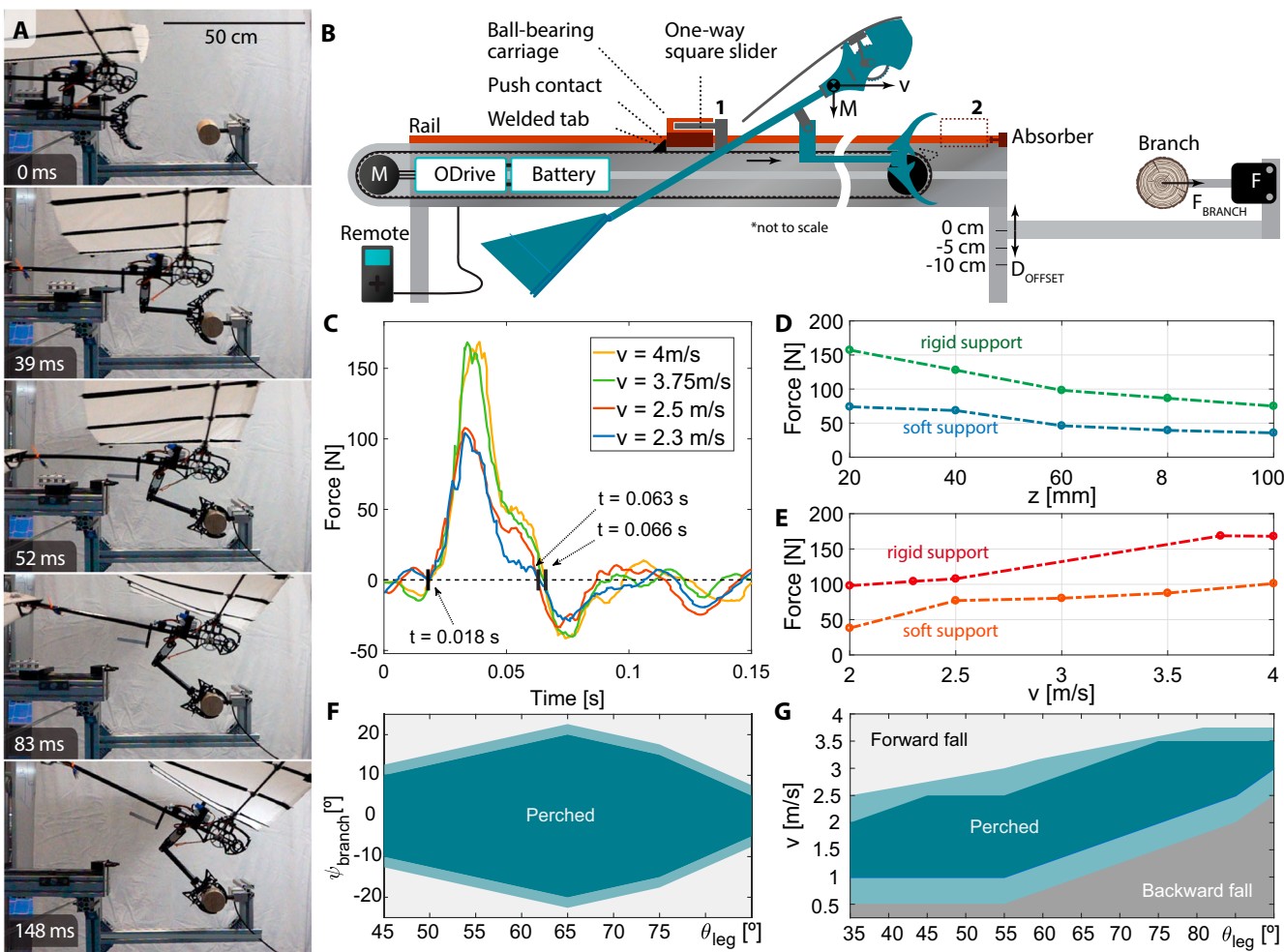

**Fig. 5 | Evaluation of perching capability. A** Time frames of the approaching leg-claw system and the test robot with similar mass to the flight version. **B** Launching experiments setup with the robot in the acceleration position[1], the released point[2], and the perching point. **C** Force measurement upon impact at different speeds. **D** Force measurement as a function of misalignment. **E** Force measures as a function of speed. **F** Perching success region considering leg angle $\theta_{leg}$ versus velocity magnitude. The perching space is divided into three regions: 1, the system falls over (light gray); 2, success in perching (blue); 3, the system falls backward (dark gray). 90° represents a horizontal leg. **G** Perching success region considering leg pitch angle $\theta_{leg}$ versus yaw-deviation $\psi_{branch}$ of the branch with respect to the velocity direction. A 0-degree deviation of the branch means that the velocity direction is completely perpendicular to the axis of the branch. The velocity magnitude is fixed at 2.5 m/s. The space is divided into success (blue) and no success (light gray).

maximum angle attainable as it cannot provide more than 5.5 Hz. Experiments show that pitch angles above 40° result in insufficient speed and consequently loss of lateral control. As such, a pitch angle of 30° is appropriate for perching flights. This set point is maintained by a proportional-integral (PI) controller. Figure 6E shows that the controller reaches the target 30° angle within 1 s, without significant overshoot and is maintained throughout the flight. Consequently, when the pitch reaches 30°, the velocity drops to 2.5–3 m/s. This velocity is maintained until perching, shown in Fig. 6F.

The altitude target value is achieved through thrust control action. Using this method, all flight trajectories reached the 2 m set-point within 8 m of flight distance, as shown in Fig. 6B. The trajectory from the motion capture system is reported in Supplementary Movie 9. At branch location, i.e. $X = 14$ m, deviation from the set-point is within ±10 cm. The lateral deviation shown in Fig. 6D is within 0.6 m, rarely exceeding the branch dimension of 0.8 (indicated in blue). The flight controller performs adequately for different branch altitudes as demonstrated in Fig. 6G, with a maximum average error of 16 cm, data available in Supplementary Table 3. The last reported flight (2nd Robot) follows a slightly different trajectory to the rest of the experiments. This occurs because this flight was performed with a second robot which inevitably shows manufacturing differences. In

particular, the second robot does not use wing camber. Nonetheless, even with those differences, perching was achieved without any modification to the flight controller and parameters, demonstrated in Supplementary Movie 10.

The variations around the reference values in $V_x$, $\psi$, $\theta$, $Y$ and $Z$ at the time of perching are small (Supplementary Table 4). The results of the six perching experiments show the following variations: $V_x \sim 2.07$ to 2.8 m/s; $\psi \sim -8.3°$ to 4°; $\theta \sim 23.6°$ to 31.8°; $Y \sim -0.23$ to 0.02 m; and $Z \sim 1.95$ to 2.06 m. In summary, the perching performance is successful around the reference values and the leg-claw mechanism with active leg control compensates for the inaccuracies and resists the impact under those conditions.

## Discussion
Through advances in flapping-wing technology, we demonstrated the first successful branch perching flight with an ornithopter. Thanks to the controlled attitude/altitude and fast tail action, the flight speed was regulated around a set-point of 2.5–3 m/s, and the position within a 15 cm-close range of the target. The proposed actively controlled claw-leg system (equipped with vision feedback) delivers close-range control, fast grasping within 25 ms and high torque of 2 Nm. The purely mechanical release action on impact

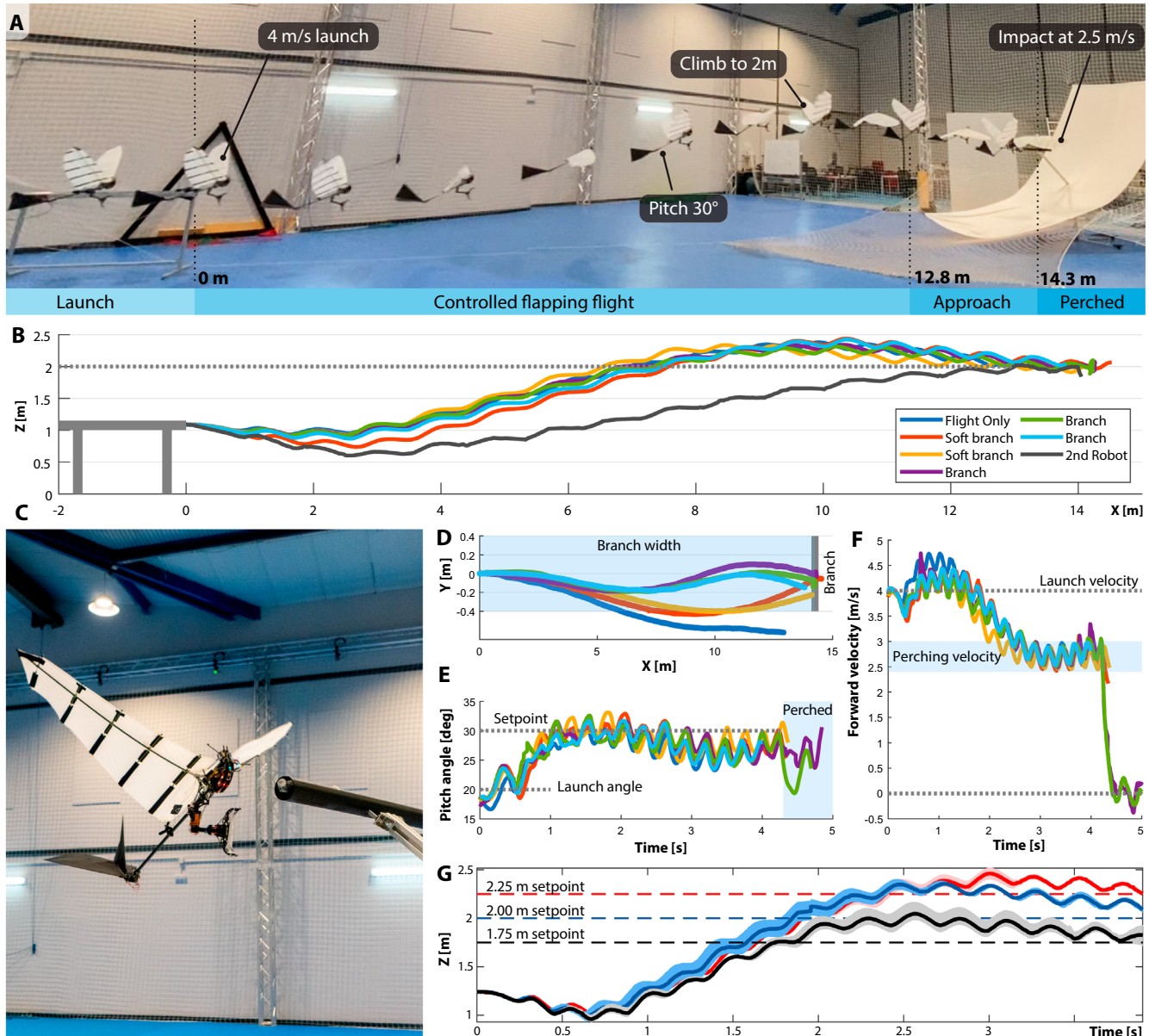

**Fig. 6 | Perching flights with a flapping-wing robot. A** Photo sequence demonstrating the flight and perch maneuver. **B** Vertical trajectory with altitude controller set-point of 2 m. After tuning the controller and line sensor, 9 attempts in total have been done for perching with 6 successful ones, resulted in 66% success rate. **C** Robot in the approach phase to the branch. **D** Top-down trajectory showing lateral deviation from the 0°. **E** Pitch behavior during flight, stabilized to 30°. **F** Forward velocity of the robot, initially at 4 m/s, was reduced to 2.5 m/s at the branch location. The velocity measures do not fall to zero on impact, due to an automatic early end of the logging. **G** Altitude of the flight experiments with different set points: blue color demonstrates flights targeting 2 m height, red color 2.25 m, and black color 1.75 m. The plots show the mean value and variance of 3 sets of flights.

guaranteed closing without external sensing or actuation, increasing perching success.

Successful full perching flights of the P-Flap were achieved through a series of flight phases, described in Fig. 6A. In the first part of the flight (launch phase), the launcher accelerates the robot to 4 m/s. Results show that launches are consistent, both in direction and speed. At the end of the rail, the robot's position and attitude are wirelessly streamed and the robot's flight computer enters a controlled flapping flight (controlled flapping-flight phase). Tracking data shows that the flight controller achieved a good regulation of the pitch angle at $\Theta = 30°$, zero yaw angle $\Psi = 0°$. When the robot gets to within 150 cm of the branch, the detection system is enabled (approach phase), see robot in position in Fig. 6C. The line sensor located on the claw feeds the relative position of the branch to the leg microcontroller. The leg

servo corrects the angle of the leg to align it with the branch. During this approach phase, the flapping motion is maintained and the flight controller is active. At 20 cm from the branch, the flapping is stopped. Immediately after hitting the branch, the claw locks onto the branch. Experiments have demonstrated that we can reach the branch location repeatedly, within the tolerance range of the leg-claw system. The low forward flight velocity shortly before landing ensures that forces remain below 150 N, significantly reducing the likelihood of damage. We report that branch perching was achieved in 6 out of 9 perching flight tests, and that no damage occurred in the successful flights. The 3 failed perching maneuvers narrowly miss the branch, and are reported in Supplementary Table 4 and Supplementary Fig. 18.

While perching has successfully been demonstrated experimentally, the conditions were controlled to simplify the process.

Specifically, that scenario simplifies the controller design, ensuring that the linear controller has enough authority to cope with small disturbances around a feasible path. Optimized bio-inspired flight paths would improve perching reliability and reduce impact forces[36]. Also, the indoor tracking zone yields accurate and fast position/orientation, which exceeds what is currently obtainable with satellite/inertial-measurement-unit (IMU) technology. Both will inevitably degrade the flight accuracy in an outdoor scenario, with disturbances and lack of measurement accuracy, resulting in more difficult perching maneuvers, and hence, more complex control algorithms will be strongly needed. Our first approach of local detection with a line-scan sensor would also be limited. New camera-based methods, which are currently being investigated, would greatly increase precision improvements in trajectory planning and close-range active control[37]. Finally, in terms of physical interaction, new materials and soft-robotic methods for the claw, such as shape-memory alloy (SMA), will further improve the conformability around a different variety of branch sizes and shapes[38]. Lastly, taking-off from the branch will be the focus of future work. While the leg-claw system can fully re-open from the branch, the absence of initial velocity currently impedes taking-off flight. Several strategies are envisioned. First, correct the posture once the system is perched to obtain a stable configuration where, with almost zero friction, the claw can re-open slowly. This would leverage the previous work addressing balancing challenges[31,39]. Then, the leg actuation should be improved to execute a jump forward, and with maximum flapping thrust. Secondly, given sufficient ground clearance, the vehicles could fall from the branch and perform a controlled flight recovery, once their speed reaches 2.5 m/s. A larger tail could further reduce this speed.

In summary, with the added perching capability, large-scale flapping-wing robots such as the P-Flap presented here can interrupt their power-intensive flight on a branch, with the primary benefit of saving energy, see power measures in Supplementary Method 7. This opens a broad range of possibilities, from local wildlife observation to sample return, as well as for other applications i.e. contact inspections and manipulation. This is possible due to the safe nature of flapping-wing locomotion, reducing the impact on animals and humans and significantly easing deployment authorization requirements.

## Methods

### Flapping-wing robot P-Flap

The P-Flap, without the perching appendage, is a 520 g flying robot, including battery, designed for research and physical interaction. The details of the components (weight distribution) are reported in Supplementary Fig. 9. Ease-of-repair and modularity are key underlying design drivers. The structure of the perching robot consists of three separable sub-systems, namely a central body, a pair of membrane wings, and a two-DoF tail. An assembly instruction manual is available in Supplementary Method 6. The body is built around a parallel set of carbon-fiber/FR4 plates. Between the plates is located a 42 : 1 transmission which reduces the 150 W brushless motor output to a 5.5 Hz rotational motion. This actuation is then transferred to the wing via a crankshaft and a four-bar linkage, producing an oscillatory motion of the wing. The discrete components, i.e. the battery, radio control (RC) receiver, electronic speed control (ESC), companion computer are all connected via a unifying PCB

Motion capture localization data is given through virtual-reality peripheral network (VRPN) and the actuator instruction is sent via a serial peripheral interface (SPI) to the low-level controller. Power on the robot is sourced from a 4S LiPo battery. The voltage is then regulated by three on-board DC–DC integrated power modules to 3.3 V, 5 V, and a variable voltage. This powers first, the microcontroller and the infrared light-emitting diodes (LEDs), secondly the companion computer, and lastly the tail actuators and leg. The variable voltage can be adjusted from 5–10 V, depending on servo requirements. See

Supplementary Method 5 and Supplementary Fig. 19 for the electronics description and schematics. The tail of the flapping-wing robot enables pitch and yaw control via two high-performance servo actuators (see Supplementary Fig. 2 for actuator comparison). The selected conventional tail configuration yields high pitch authority, an important metric for precise vertical positioning[40]. Low weight and extended pitch deflection are possible using the new direct drive, carbon-fiber tail design, see Supplementary Fig. 14. The CAD of the perching robot with leg-claw system is accessible online and in the available data, complemented by the list of components and their source in Supplementary Table 5.

### Launcher setup

To investigate impacts, flight, and perching maneuvers, we employ a launching apparatus to obtain repeatable initial conditions and fast transition to flight. This system features a 2 m double rail with a ball-bearing carriage, see a 3D assembly view in Supplementary Fig. 10. The carriage is accelerated by a push-tab on the timing belt, which allows the belt and carriage to take off at the end of the rail. At that point, the carriage's energy is absorbed by dampers while the belt and motor's energy is injected into a braking resistor. The operation of the system is handled via a handheld Arduino remote which sends the instructions to the brushless motor driver. The trajectory is computed to linearly accelerate the robot via a 2.3 kW motor to a user-set final velocity of 4 m/s on a 1.6 m length. The launcher imparts 5.6 J to the robot at separation. The robot is held at its trailing edge via a sliding adapter with rotation locking, details in Supplementary Fig. 11. The whole robot-sliding adapter is placed at a lateral offset of 40 cm to avoid any collision of wings or tails with the launcher structure. The analytical model and CAD of the launching system are accessible online and in the available data.

### Flight experiment setup

All the flights were performed in the 20 × 14.5 × 8 m flight space fitted with 28 Opti-track infrared (IR) cameras. The complete setup is shown in Supplementary Fig. 8. The launcher system is placed in one corner, taking advantage of the longer flight distance of a diagonal trajectory. The $X$ axis is aligned diagonally. Flight control and flapping start as soon as the tracked robot moves past the end of the launcher, marking the zero position.

### Leg-Claw system manufacturing

The claw system consists of four identical claw plates, CNC-cut in 1.5 mm-thickness cured carbon fiber. Joined by 20 mm spacers, they form the top and bottom claw. These two parts pivot around the four extremities of two outer carbon-fiber plates, which link the assembly to the leg. An energy-storing spring of 11 mm diameter fits between the claw plates. The 40 mm-long tension spring applies a force of 111 N with a 5 N/mm rate of extension and weighs 8 g. This choice takes into account the weight budget, and maximum allowable diameter to fit within the plates and is limited by the maximum force of the re-opening actuation (Supplementary Method 1 for more details). Thanks to the four pivot points, the spring is free to pass the pivoting center. Fitted on each of the claw plates is a 3D-printed insert and spike holder (Supplementary Fig. 12). The cast Ecoflex membrane spans between the four inserts, covering the spring. The branch detection is performed using the signal obtained from a TSL1401 line-scan sensor (1 ×128 pixels). Typically used in the barcode scanning device, they are well-suited for detecting a horizontal line in controlled settings. Due to the low amount of data produced by this sensor, the signal readout is fast, up to 4 kHz given sufficient light in the scene. In practice, indoor lighting conditions limit this rate to 200 Hz for adequate signal-to-noise ratios. More importantly with these sensors, the data can be analyzed on a low-powered microcontroller, reducing weight and power requirements. While the

selected vision system performs well in a controlled environment, its low visual acuity (28 arcmin/px) limits the branch detection range to 7.7 m in theory, but only 2 m for reliable experimental detection, a value acceptable for this perching task. This could be improved with a higher resolution line sensor, e.g., the epc910 or 4 K camera modules, achieving 1.7 arcmin/px in the Skydio 2. Predatory birds are capable of even higher visual acuity, with the American Kestrel distinguishing elements as small as 0.4 arcmin. The sensor and the resulting actuation signal are reported in Supplementary Fig. 5. The signal is read at 330 Hz and processed by a Seeeduino Xiao 48 Mhz microcontroller, one of the smallest commercial solutions available at 20 × 17 mm. It is connected via I2C protocol to the flight companion computer. The leg itself runs a state machine, permitting different operation modes: fixed-angle, closed-loop control, claw re-opening, and data return. The signal is rectified with a cosine function and a dynamic threshold is calculated from the average brightness of each frame. The highest-located dark line is accepted as the branch. The leg features a carbon plate construction and hip joint based around a BMS 922+ servo (assembly in Supplementary Fig. 13). This actuator was selected for the highest torque-to-weight ratio within a 15–20 kg cm torque range, by using the relation illustrated in Supplementary Fig. 2. The claw is compression-fitted at the extremity of the leg structure. The whole leg-claw appendage is affixed to the robot's body tube via a compression fit clearance, allowing for position and angular adjustment.

## Data availability

The source data required to validate the experiments, construct a similar flapping-wing robot, and verify the reported results generated in this study have been deposited in the Zenodo database under accession code https://doi.org/10.5281/zenodo.7225970. This data folder is complemented by the supplementary information files.

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

## Acknowledgements

Our sincere thanks for all the support from our fellow group members. In particular, we would like to thank V. Perez Sanchez for his work on the elastomers and his help in various experiments. Thanks to F. Maldonado Fernandez for his work helping with the flight control system and to M. M. Guzman Garcia for her work on prototype developments. Thanks to A. Satué Crespo for his contribution to the launcher experiments. Thanks also to S. Pellegrini for his work on the re-opening mechanism. Thank you to D. Garcia Marti for her generous support, ideas and comments on the manuscript. The research is funded by the European Project GRIFFIN ERC Advanced Grant 2017, Action 788247 (all authors). This project has also received part funding from the European Union's Horizon 2020 research and innovation program under the Marie Skłodowska-Curie grant agreement No 101029670 (R.Z.).

## Author contributions

Conceptualization: R.Z., D.F.T., S.R.N., J.A.A., A.O. Design and development: R.Z., J.T.B., D.F.T., S.R.N., J.A.A. Experimentation: R.Z., J.T.B., D.F.T., S.R.N. Flight control: D.F.T., S.R.N., J.A.A. Data curation: R.Z., D.F.T., S.R.N., J.A.A. Illustrations and media: R.Z., J.T.B., D.F.T., S.R.N. Funding acquisition: A.O. Project administration: A.O. Supervision: R.Z., J.A.A., A.O. Writing – original draft: R.Z., D.F.T., SRN Writing – review & editing: R.Z., D.F.T., S.R.N., J.A.A., A.O.

## Competing interests

The authors declare no competing interests.
