## [Peer Review File · Nature Communications]

REVIEWER COMMENTS

Reviewer #1 (Remarks to the Author):

In this paper, the authors described a novel perching demonstration that is performed by a 700 g ornithopter robot. Overall, the authors have done a tremendous amount of work to extensively characterize perching performance. In my view, perching becomes difficult as the robot size grows, and this is an impressive result for a robot at this scale. The proposed work is relevant for researchers in this field. My major and minor comments are described below.

Major comments:

1. The title of this paper is “How ornithopters can perch autonomously on a branch”. It gives the impression that the entire maneuver is autonomous. In the main text, the authors describe that while computation and control is done onboard, motion tracking is done offboard. The perching maneuver strongly depends on accurate position feedback while vision-based approach alone is insufficient. The authors should clearly define what type of “autonomy” that they are referring to in the paper.
2. Based on Figure 6, the authors showed a series of perching flight results. The authors show 3 repeating flights as well as a flight performed by a different robot. I am curious about perching success rate and robustness. How many perching flight tests have been conducted and out of those, how many of them are successful? In addition, it seems all flights are conducted where the branch is perpendicularly positioned relative to the flight path. Can the branch be positioned at an angle?
3. One primary use of perching is to save energy. Can the authors comment on that amount of energy that is required to perform a perching demonstration? During robot flight, how much more energy is required to carry the claws and related actuators?
4. In the introduction, the authors acknowledged that prior works have demonstrated perching and takeoff of fixed-wing and rotary wing aerial robots. What is particularly challenging about the perching of flapping-wing robots? If I applied either micro spines or bird-like claws as described in existing works, will these designs work for your robots? Is the major contribution of this paper related to enabling perching for an ornithopter robot or a macroscale (700 g) robot?

Minor comments:

1. In the discussion section, I hope the authors can discuss about future strategies on takeoff from a branch. The current design needs 20 s to open the claw, which maybe too slow to enable takeoff. In addition, what is the minimum required takeoff speed. For a 700 g ornithopter robot, what is the amount of energy needed from the catapult?
2. In the supplement, some equations are numbered but some are not.
3. Some font sizes in the supplementary figures are not consistent (e.g., Fig. S9 and Fig. S10)

4. Page 12 of main text, missing a space in the second to last line

Reviewer #2 (Remarks to the Author):

The authors present the first autonomous perching flights of a flapping-wing robot on a branch. The main contribution is the design of a bi-stable leg-and-claw system that can grasp a branch in a matter of milliseconds with the help of pre-tensioning. The full control loop is closed, with the flapping wing robot approaching the branch and keeping its height with the help of a motion-tracking system and performing adjustments of the claw position based on a linear camera in a high-contrast environment (black branch, white background). Seven successful flights, four of which with actual perching, are presented.

The article presents a significant advance in the domain of flying robotics, as perching will be a very useful capability for any long-term autonomous flying system. Other work in this burgeoning area of research is typically performed with quadrotors. As is well explained by the authors, perching with a flapping wing, especially one that is not able to hover, is more challenging due to the flapping wing dynamics. The article is in general very well-written, and is illustrated with beautiful and clear illustrations. Especially the claw and grasping system has been studied in a lot of detail, as befits an article for a top journal as Nature Communications.

I do have a number of remarks and concerns for which I would like the authors' reply, and which may lead to improvement of the article.

Main remarks:

1. In Fig. 6, seven flights are shown, one for flight only, one with a second robot, two with a soft branch (in which the claw does not close as is evident from the video), and three with a hard branch. The videos show two flights with the hard branch (movie S7), not three. Moreover, the videos are cut off when the claw has attached, not when the robot is completely at rest (the slow-motion one does stop at that point, showing that the robot does not fall forward or backward). As a roboticist myself, I know how involved robot experiments can be, but together, this raises the question how repeatable the flights are, and whether the robot indeed remains at rest after perching. Why was only one flight performed with the second robot? Have there been no failures in flight? Failed flights would not form a show-stopper in my eyes, but rather a learning opportunity. On this matter, on

page 15, the authors state: “We report that branch perching was achieved in 6 out of 9 perching flight tests, and that no damage occurred in the successful flights.” Besides the fact that the numbers are confusing, the 3 failed flights are not analyzed anywhere, as far as I can see. Working with the numbers shown in Fig 6, in my eyes, seven flights suffice to show that the motion tracking system allows the robot to reach the branch, but there are very few actual perching maneuvers (four). Ideally, more flights would be performed, or, if really not possible, a thorough discussion of reliability should be added.

2. The authors state in the introduction: “We also introduce a **development process**, consisting of four experimental steps which lead to autonomous perching flights.” The stars indicate bold text. This sounds as if this development process is a major contribution of the article. Looking at the development process, illustrated in Fig. 4, it is very sensible, but it does not seem so innovative that it merits mentioning as a major contribution. I suggest that the authors slightly downplay this particular contribution in the introduction.

3. In the discussion, the authors evaluate the use of the re-opening for taking off again: “Then, the leg actuation should be improved to execute a jump forward, synchronized with claw re-opening. Secondly, given sufficient ground clearance, the vehicles could fall from the branch and perform a controlled flight recovery.” However, in the same time, opening takes around 20 seconds... How can a jump be synchronized with such a slow opening?

4. My compliments for Fig 1. It is beautiful. I do think it would be even more illustrative to indicate the size and weight of the flapping wing robot in some way (perhaps simply with text for wingspan and total mass), add an motion tracking camera to the image, and adapt “last-meter sensor” to something like “linear vision camer for last-meter corrections”.

Minor remarks:

1. “design can lock” -> “design that can lock”
2. Ref 6 seems to miss a title and venue.
3. “The unsteady aerodynamics of the vehicles lead to hard-to-model dynamics and therefore inaccurate positioning” -> Is the flapping in normal flight conditions not quite stereotypical? Or do you see large variations in the flapping amplitude in nominal flight conditions? I think it can be said that the flapping motion leads to additional disturbances and less accurate control.
4. “The large size of ornithopters further increases the perching difficulty.” -> The definition of ornithopter is an aircraft that flies by flapping its wings (<https://en.wikipedia.org/wiki/Ornithopter>). This does not have any consequence for its size (ornithopters can be as small as Harvard’s robobee). Please be more specific here, e.g.: “Moreover, perching is more difficult for large (> 1m wing span)

ornithopters.” Similarly on page 10: “While ornithopters are passively stable, it” -> tailed ornithopters are typically passively stable, tailless ones are not.

5. The authors state on smaller flapping wing robots: “(8, 9), however they suffer from limited payload.” Later, they state on larger flapping wing robots: “Flapping-wing robots also face additional constraints such as stringent payload restrictions and oscillations that need to be addressed.” This forms a bit of a paradox... Is the flapping-wing robot size selected by the authors the “sweet spot”, where payload restrictions are interesting and not limiting? Or can the current work lead to insights that will help also smaller flapping wing robots to perch? As I interpret it, the authors work with a larger system, since it is currently easier to add the necessary hardware for perching. A more detailed / subtle reflection would be appreciated.

6. How do the perching multirotors compare to the current platform in terms of size and mass?

7. “The geometry of the claw is modeled to offer maximum friction at the branch level while having a large opening angle and low trigger force.” -> Is this provably the maximum level of friction?

8. “The claw dimension is smaller than the flight position accuracy” What do you mean here? How was the flight position accuracy measured / determined?

9. Page 8: “Future outdoor perching maneuvers in cluttered environments will benefit from improved sensing methods, e.g. full 2D or 3D vision systems (22).” -> The authors could refer here to work with event or CMOS cameras on flapping wings here as well:

a. Eguíluz, A. G., Rodríguez-Gómez, J. P., Paneque, J. L., Grau, P., de Dios, J. M., & Ollero, A. (2019, November). Towards flapping wing robot visual perception: Opportunities and challenges. In 2019 Workshop on Research, Education and Development of Unmanned Aerial Systems (RED UAS) (pp. 335-343). IEEE.

b. De Wagter, C., Tijmons, S., Remes, B. D., & de Croon, G. C. (2014, May). Autonomous flight of a 20-gram flapping wing mav with a 4-gram onboard stereo vision system. In 2014 IEEE International Conference on Robotics and Automation (ICRA) (pp. 4982-4987). IEEE.

10. “Selection of a suitable trajectory in which no control limitations appear, such as saturation or limited stability margins” -> What trajectories are / are not possible and why?

11. Page 10: On one hand / on the other hand: Are these really opposing things?

12. Page 11: “up to 5m/s” Where can we see that? Fig 5E goes up to 4 m/s.

13. “results show that the system can tolerate a branch deviation in a range of ± 10 - 20° depending on the leg pitch orientation” -> Is this not about yaw?

Nature Communications 2022 - Response to reviewers

Manuscript **NCOMMS-22-28797**

How ornithopters can perch autonomously on a branch

September 2022

We would like to thank both reviewers for their insightful and constructive comments. Our first submission featured shortcomings in terms of analysis and discussion. We have put our full efforts to improve the manuscript based on the excellent comments. We hope that the reviewers will find our answers adequate and would welcome any additional feedback. This document combines both comments and answers of the two reviewers.

As instructed, we have created a data repository on Zenodo containing flight trajectories, launcher measures, design files, sizing calculations, embedded software and launcher design files. This, in addition to the supplementary material descriptions, will enable the reader to validate the experiments, construct a similar flapping-wing robot, and verify the reported results.

The answers are formatted with an offset and color, while text changes are mentioned as follow:

“this quoted text has been changed or ~~removed~~.”

1 Reviewer: 1

In this paper, the authors described a novel perching demonstration that is performed by a 700 g ornithopter robot. Overall, the authors have done a tremendous amount of work to extensively characterize perching performance. In my view, perching becomes difficult as the robot size grows, and this is an impressive result for a robot at this scale. The proposed work is relevant for researchers in this field. My major and minor comments are described below.

1.1 Major comments

Comment 1 *The title of this paper is “How ornithopters can perch autonomously on a branch”. It gives the impression that the entire maneuver is autonomous. In the main text, the authors describe that while computation and control is done onboard, motion tracking is done offboard. The perching maneuver strongly depends on accurate position feedback while vision-based approach alone is insufficient. The authors should clearly define what type of “autonomy” that they are referring to in the paper.*

We fully agree with the reviewer that “autonomy” can be interpreted in different ways. In this context, as accurately pointed out, the perching flight was autonomous in the sense that no manual input was given during the entirety of the maneuver, and all computing done onboard - yet precise position feedback was available. We clarify this in the introduction as suggested:

“In this paper we propose a novel autonomous perching method capable of landing and maintaining large flapping-wing robots on a branch. This three-phased method consists in the simultaneous operation, within a flying vehicle, of a flapping-flight controller, a close-range correction system and a passive perching appendage. The perching method is validated with a 1.5 wingspan - 700 g robot, depicted in Fig. 1-A, resulting in the first indoor autonomous branch perching demonstration of a large flapping-wing robot. The flight-perch maneuver is performed autonomously with localization from a motion capture

system, ending on a wooden branch at a 2 m altitude, see a time-lapse in Fig. 1-C. The level of autonomy achieved in this study is without user input during the entirety of the experiment, fully on-board computation and navigation, but dependent on accurate localization data from a motion capture system.”

Comment 2 *Based on Figure 6, the authors showed a series of perching flight results. The authors show 3 repeating flights as well as a flight performed by a different robot. I am curious about perching success rate and robustness. How many perching flight tests have been conducted and out of those, how many of them are successful? In addition, it seems all flights are conducted where the branch is perpendicularly positioned relative to the flight path. Can the branch be positioned at an angle?*

Over 80 flights have been performed, before reaching the perching experiment stage. The mechanical structure of the bird has been designed to reduce the weight as much as possible for having room for additional equipment, as a result, unsuccessful perching maneuvers are likely to cause damage. This is the why we gradually increased the “damage likelihood” by moving from launcher experiments, to flight experiments, to soft branch experiments and finally full wooden branch experiments. When the flapping bird was finally optimized for perching, we managed the maneuver and record the presented results in this paper with 9 flights, out of which 3 missed the branch. The robustness of flight depends on many factors, all of them should work precisely to perch successfully: onboard controller, motion capture feedback, line sensor for the branch detection, and leg controller.

So, in terms of robustness, with everything is tuned, the success rate is approximately 66%. The success rate has been added to the caption of Fig. 6 (in the revised manuscript). In addition, extensive flight characterization was performed subsequently (this time without a branch), and is reported in Fig. 6-G (in the revised manuscript) for different set-points, and also in another recent publication (Figure 8 of Ref. [R1] reply-to-comments letter, below):

[R1] S. R. Nekoo, D. Feliu-Talegon, J. A. Acosta and A. Ollero, “A 79.7g Manipulator Prototype for E-Flap Robot: A Plucking-Leaf Application,” in IEEE Access, vol. 10, pp. 65300-65308, 2022, doi: 10.1109/ACCESS.2022.3184110.

We have analyzed the results of the 3 failed flights, see below Table 1. We notice that in Fail (I) and Fail (II), the variables V_x , Yaw angle, Pitch angle and $Z(m)$ are in the range of the successful perching, however, the variable Y is very high, so the lateral control is not accurate enough and the ornithopter is not able to reach the desired branch. Figure 1 (reply-to-comments letter, below) shows the deviation in the lateral movement in comparison with the 4 experiments that are successful using the rigid branch. On the contrary, in the experiment Fail (III), all the variables, except for Z , are in the admissible range of values of the successful perching experiments. However, the system is not able to grasp the branch. This is explained because the variable Z is out of the range of the successful perching flights. Then, the system reaches the X and Y position of the branch with a very high altitude. In this experiment the claw grazed the branch but does not grasp it. The ornithopter subsequently fell into the net after the branch. The control of altitude was not accurate enough and the actuated leg was not able to compensate the misalignment between the center of the branch and the claw. Figure 2 (reply-to-comments letter, below) shows the altitude movement of different flights showing that in the experiment Failed (III), the variable Z is the highest of all at the position of perching.

These experiments show the importance of having a very accurate control of all the variables of the system and the difficulty of succeeding in the perching of flapping wing robots.

Table 1: Analysis of failed flight (added to Table S5)

	V_x [m/s]	Yaw angle [°]	Pitch angle [°]	Y [m]	Z [m]
Fail (I)	2.38	0.473	28.22	-0.523	1.992
Fail (II)	2.22	1.67	26.89	-0.518	2.06
Fail (III)	2.45	-0.1	28.23	-0.231	2.09

Figure 1: Analysis of failed flight.

Figure 2: Analysis of failed flight.

The analysis and figures have been added in the supplementary material as table S4 and Fig S17. A mention in the main text has also been made:

“We report that branch perching was achieved in 6 out of 9 perching flight tests, and that no damage occurred in the successful flights. The 3 failed perching maneuvers narrowly miss the branch, and are reported in Table S4 and Fig. S18.”

Regarding whether the branch can be positioned at an angle, we performed perching tests (see manuscript Fig 5-G) with the branch rotated in yaw, which we understand is what the reviewer is asking. The system was tolerant to deviations of up to 20°. The current claw design is relatively narrow (20 mm distance between plates), limiting how well it can resist to angled perches. A second leg or a wider single claw

would certainly improve this aspect, as the expense of added weight. These tests were done with a branch positioned shortly in front of the launcher (and therefore not in across-the-room flights), however we expect the same behavior to happen in full flights..

Comment 3 *One primary use of perching is to save energy. Can the authors comment on that amount of energy that is required to perform a perching demonstration? During robot flight, how much more energy is required to carry the claws and related actuators?*

We would like to thank the reviewer for this insightful comment which adds breath to this research. We have performed an analysis with experimental data from on-board power measures, explained hereunder.

The robots' power consumption in flight strongly depends on the flapping frequency as shown in Table 2. Extrapolating from in-flight measures with curve fitting (curve equation is $P = 2.66f^{2.27}$) gives a power consumption of 61.8W at 4Hz flapping frequency. This is the frequency that has been observed experimentally in the perching level flight sections.

Table 2: Power measurements collected onboard during active free flight. Note: the robot's mass in this test is 670g, or 4% lighter than in the perching demonstration.

	Frequency	Current	Power
Measured	2.9 Hz	1.8 A	29.7 W
	3.5 Hz	2.8 A	46.2 W
	4.7 Hz	5.2 A	85.8 W
	5 Hz	6.4 A	105.6 W
Extrapolated	4 Hz		61.8 W

From the total power measurements at the battery level, we need to subtract the constant, flight independent power consumption of the onboard avionics. A breakdown of the power draw of individual subsystems is presented in Table 3.

Table 3: Avionics power draw

System	Power measurements
PCB	0.81 W
Nanopi companion computer	0.69 W
Current sense	0.18 W
RC receiver	0.41 W
ESC (brushless motor driver)	0.41 W
Optitrack infrared LED	0.99 W
Tail servos (rudder/elevator)	1.6-5.1 W
Avionics total	6.8 ± 1.8 W

Taking the propulsive power of the 700g flapping vehicle in level flight as $61.8W - 6.8W = 55W$, we scale this value for a lower weight as follow.

The power P in cruising conditions is a product of the thrust T and the cruise velocity V , that is, $P = TV = DV$, where the thrust is balanced by the drag D , and the lift is balanced by the weight. For a given lift-to-drag ratio, the thrust is proportional to the weight: $T \propto W$, so $P \propto WV$. In cruising conditions, $V \propto W^{\frac{1}{6}}$, and therefore $P \propto W^{\frac{7}{6}}$ [R2] (reply-to-comments letter, below).

According to these scaling estimates, the resulting propulsive power required for a robot without leg and claw (which would weigh 516g) would be $39W$, or a saving of $16W$. This is equivalent to a 26% power cost for the additional perching appendage. Given a 4 second flight maneuver, the required energy expenditure is summed up in Table. 4.

Table 4: Flight energy

Scenario	Duration	Energy estimate	Battery usage
Perching flight	4 s	248 J	0.93 %
Claw loading	20 s	60 J	0.22 %
Perched (servos off)	1 h	7550 J	28.00 %
Flight without claw-leg	4 s	182 J	0.68 %
Battery capacity		26700 J	100.00 %

[R2] Liu, Tianshu. "Comparative scaling of flapping-and fixed-wing flyers." AIAA journal 44.1 (2006): 24-33.

The measures, calculations and results above have been added as supplementary material Text S7. A mention to this new material is made in the main text as follow:

"In summary, with the added perching capability, large-scale flapping-wing robots such as the P-Flap presented here can interrupt their power-intensive flight on a branch, with the primary benefit of saving energy, see power measures in Text S7."

Comment 4 *In the introduction, the authors acknowledged that prior works have demonstrated perching and takeoff of fixed-wing and rotary wing aerial robots. What is particularly challenging about the perching of flapping-wing robots? If I applied either micro spines or bird-like claws as described in existing works, will these designs work for your robots? Is the major contribution of this paper related to enabling perching for an ornithopter robot or a macroscale (700 g) robot?*

Thank you for this comment. We believe that fixed-wings and multirotors fall into quite distinct categories when it comes to perching maneuvers for the following reasons:

	Multirotors	Fixed-wing	Flapping-wing
Hover	Yes	No	No
Oscillations	No	No	Yes
Fluid-structure interaction	No	No	Yes

Table 5: Challenges of flapping-flight for perching

The first aspect, hover capability, removes the need to **resist the impact** stemming from the forward flight velocity of winged drones. Importantly, drone perchings are executed from above thus hardly requiring any **unbalance** resistance. The maneuvers can happen as **slow** as needed, and trajectory planning can be time-independent. Lastly, the **position precision** is vastly superior to that of a forward flying robot, on the order of millimeters in motion capture systems. As such, landing a multirotor on a point is more accessible.

The second aspect relates to the oscillations in state variables, which are inherent to flapping-flight. This further degrades the position precision of the trajectory. Neither fixed-wing vehicles nor drones suffer from this issue.

The third aspect is fluid-structure interactions. A flapping wing significantly deforms in flight which makes modeling very challenging. Even after decades of flapping-wing research, there is still no modelling framework that accurately captures the complex interactions around flapping wings. This makes it challenging to design and control these robots precisely.

We would like to add that to this day, neither fixed-wing robots nor flapping robots have been shown to perch on a branch, even if excellent fixed-wing attempts have been made recently [R3] (reply-to-comments letter, below). The first aspect is certainly the most problematic.

If we were to add a **micro-spine** system to our robots, we envision several issues. First, we would likely miss the branch with the micro-spine, as the current claw does open much wider than the branch, providing tolerance to misalignment. Even if the spines would hit the exact middle of the branch, they would likely break, bend or detach from the branch due to unbalance. Indeed, in the cited works, perching occurs on surfaces, which enables vertical separation between contact points. This is not possible on a branch.

Regarding the **other claw systems**, there is certainly excellent designs that could potentially be adapted to our robots. However, impact-resistance has not been demonstrated at our forward flight speed. Robustness is also another challenging aspect as designs heavily relying on 3D-printed elements might suffer from faster degradation. In contrast, our composite construction has withstood every single experiment, from launcher tests to failed flight without failure. Lastly, the weight of the appendage is critical to reduce impact energy, reduce flapping frequency and improve flight performance. To the best of our knowledge, our leg-claw design is the lightest for a 700 g robot, even when including the optical vision system.

Very recently (July 2022), researchers have further emphasized that perching with flapping robots are of high interest, aiming to achieve this same task with smaller robots [R4] (reply-to-comments letter, below). Flight with a flapping robot has yet to be demonstrated.

We have updated the manuscript based on the information above:

"In addition, ornithopters require an impact-resilient leg-claw system capable of stopping the fast-moving vehicle and tolerant to flight oscillations. Drone perching tasks are performed from above thus not requiring significant unbalance resistance. The maneuver can happen as slow as needed and the position precision is superior to that of a forward flying robot."

"Compared to fixed-wing robots, flapping-wing robots face additional constraints such as increased payload restrictions and oscillations that need to be addressed. Overall, physical interaction and specifically perching is generating strong interest in robotics, highlighted by excellent recent research attempts (19)."

We agree with the reviewer, the major contribution of this work is the integration of all the subsystems to perform perching with a large-scale flapping-wing robot. Individual works on trajectories, flapping-wing robots, and claws for perching are currently being researched, our contribution lays above all in the integration of all the mentioned systems. So far, no research on this combined perching maneuver with flapping wing systems has been reported in the literature.

[R3] Stewart, W., Guarino, L., Piskarev, Y. and Floreano, D. (2022), Passive Perching with Energy Storage for Winged Aerial Robots. Adv. Intell. Syst. 2100150. <https://doi.org/10.1002/aisy.202100150>

[R4] K. C. V. Broers and S. F. Armanini, "Design and Testing of a Bioinspired Lightweight Perching Mechanism for Flapping-Wing MAVs Using Soft Grippers," in IEEE Robotics and Automation Letters, vol. 7, no. 3, pp. 7526-7533, July 2022, doi: 10.1109/LRA.2022.3184447.

1.2 Minor comments

Comment 1 *In the discussion section, I hope the authors can discuss about future strategies on takeoff from a branch. The current design needs 20 s to open the claw, which maybe too slow to enable takeoff. In addition, what is the minimum required takeoff speed. For a 700 g ornithopter robot, what is the amount of energy needed from the catapult?*

Thank you for this comment. We fully agree that takeoff from the branch is an important consideration for future work. We have already briefly covered the envisioned strategies in the 3rd paragraph of the discussion as follows, but have further expanded on this aspect (also recommended by Reviewer 2)

In previous works, we have explored the possibility to change the desired posture of the system once the system is perched (see references (31), (39) in the revised paper). We have in mind to use these works in the take-off approach. The idea is to change the position of the system in order to obtain a stable configuration where, with almost zero friction, the ornithopter remains perched in the branch. Once this posture is achieved, the claw starts to open slowly, decreasing the friction between the claw and the branch. In the very last second, where the claw is almost opened and the friction is very low, we propose two approaches:

1. Move the system in order to get out of its stable position, fall and perform a controlled flight recovery (only if there is sufficient distance between the branch and the ground)
2. Execute a jump forward and start flapping the wings at maximum speed to recover the flight. For this second approach, the claw-leg mechanism has to be changed in many aspects while for the first one, the claw can be used as it is

The main text has been clarified to include the answer to this question as follow:

“Lastly, take-off from the branch will be the focus of future work. While the leg-claw system can fully re-open from the branch, the absence of initial velocity currently impedes taking-off flight. Several strategies are envisioned. First, achieving the desired correct the posture once the system is perched to facilitate obtain a stable configuration where, with almost zero friction, the claw can re-open slowly. This would leverage the previous work addressing balancing challenges in (31, 39). Then, the leg actuation should be improved to execute a jump forward, synchronized with claw re-opening, and with maximum flapping thrust.”

The minimum take-off speed is 2.5 m/s. While the robots could technically fly somewhat slower (with a higher angle of attack), the tail does not provide sufficient control authority. An redesigned, larger tail could certainly improve this aspect.

“Secondly, given sufficient ground clearance, the vehicles could fall from the branch and perform a controlled flight recovery, once their speed reaches 2.5 m/s. A larger tail could further reduce this speed.”

The energy given to the robot by the launcher system is $E_{kin} = \frac{1}{2}mv^2 = 5.6J$, to reach 4m/s. Flight is technically possible already at 2.5 m/s with high angles of attack, implying that 2.2J would be sufficient. This is well below the typical jumping energy of a guinea fowl, reaching 7J for a 1420g bird [R5] (in reply-to-comments letter, below).

[R5] Parslew, B., Sivalingam, G. and Crowther, W., 2018. A dynamics and stability framework for avian jumping take-off. Royal Society open science, 5(10), p.181544.

“The trajectory is computed to linearly accelerate the robot via a 2.3 kW motor to a user-set final velocity of 4 m/s on a 1.6 m length. The launcher imparts 5.6 J to the robot at separation..”

Comment 2 *In the supplement, some equations are numbered but some are not.*

This has been addressed in the revised document.

Comment 3 *Some font sizes in the supplementary figures are not consistent (e.g., Fig. S9 and Fig. S10)*

This has been addressed in the revised document.

Comment 4 *Page 12 of main text, missing a space in the second to last line*

Space has been added

2 Reviewer: 2

The authors present the first autonomous perching flights of a flapping-wing robot on a branch. The main contribution is the design of a bi-stable leg-and-claw system that can grasp a branch in a matter of milliseconds with the help of pre-tensioning. The full control loop is closed, with the flapping wing robot approaching the branch and keeping its height with the help of a motion-tracking system and performing adjustments of the claw position based on a linear camera in a high-contrast environment (black branch, white background). Seven successful flights, four of which with actual perching, are presented.

The article presents a significant advance in the domain of flying robotics, as perching will be a very useful capability for any long-term autonomous flying system. Other work in this burgeoning area of research is typically performed with quadrotors. As is well explained by the authors, perching with a flapping wing, especially one that is not able to hover, is more challenging due to the flapping wing dynamics. The article is in general very well-written, and is illustrated with beautiful and clear illustrations. Especially the claw and grasping system has been studied in a lot of detail, as befits an article for a top journal as Nature Communications.

I do have a number of remarks and concerns for which I would like the authors' reply, and which may lead to improvement of the article.

2.1 Main comments

Comment 1 *In Fig. 6, seven flights are shown, one for flight only, one with a second robot, two with a soft branch (in which the claw does not close as is evident from the video), and three with a hard branch. The videos show two flights with the hard branch (movie S7), not three. Moreover, the videos are cut off when the claw has attached, not when the robot is completely at rest (the slow-motion one does stop at that point, showing that the robot does not fall forward or backward). As a roboticist myself, I know how involved robot experiments can be, but together, this raises the question how repeatable the flights are, and whether the robot indeed remains at rest after perching. Why was only one flight performed with the second robot? Have there been no failures in flight? Failed flights would not form a show-stopper in my eyes, but rather a learning opportunity. On this matter, on page 15, the authors state: "We report that branch perching was achieved in 6 out of 9 perching flight tests, and that no damage occurred in the successful flights." Besides the fact that the numbers are confusing, the 3 failed flights are not analyzed anywhere, as far as I can see. Working with the numbers shown in Fig 6, in my eyes, seven flights suffice to show that the motion tracking system allows the robot to reach the branch, but there are very few actual perching maneuvers (four). Ideally, more flights would be performed, or, if really not possible, a thorough discussion of reliability should be added.*

We thank the reviewer for the constructive comment. The reviewer raised a very interesting point and we agree that it is also important to analyze the failed flights.

First of all, we want to highlight that during this research many flights (> 80) had been carried out before the stage of recording the 9 final experiments. During these previous flights, we solved issues in hardware and software related that often resulted in failure and impacts as well as high amounts of parameter tuning. The robustness of flight depends on many factors, all of them need to work precisely to succeed in the perching maneuver: onboard controller, motion capture feedback, line sensor for the branch detection, and leg controller. Once the controllers of the system were completely tuned for perching, we performed the last 9 perching experiments - without re-tuning of the controller gains.

We understand that the reviewer would like to see more perching flights, which we agree would slightly strengthen the statistical results. The objective of the presented results was to demonstrate that the method is functional and that indeed flapping-wing robots can perch, but still at low TRL and with caveats (controlled conditions, motion tracking, known branch location etc.). We think that improvements to our method will therefore be shown in the near future, with higher reliability and better analysis in more realistic conditions.

Regarding the second robot, we carried out only one perching experiment as the purpose was to demonstrate that the method could be transferred to other another robot of similar size, and also to show that the onboard electronics, printed on the body of the bird, successfully works. We performed this single experiment and it was successful effectively showing the aspect we wanted to demonstrate.

The results of the successful flights were shown in Table 6. This analysis verifies quantitatively the performance of the integrated system, the benefits of the proposed approach and the achievements of our approach. Statistical analysis are presented through the mean, the standard deviation, the reference value and the worst value of each experiment. The results are discussed as follows:

- The little variations around the reference value in the forward velocity V_x demonstrates that the control of the speed at the time of perching is quite good taking into account the limitations in controlling large flapping wing robots. The reference value at the time of perching is 2.5 m/s whereas the variations in the speed during the experiments at the time of perching are between 2 m/s and 2.8 m/s. The results demonstrate that the system works quite well although the perching velocity varies in this range. One of the difficulties of performing perching maneuvers with flapping wing robots is that you can not control easily the speed at the time of perching as it is does with multirotor systems. Also, these results demonstrate that the mechanism resists the impact at different speeds.
- The little variations around the reference value in the yaw angle ψ demonstrates that the control loop that controls this state works well. Moreover, it shows that although the flapping wing robot is not completely perpendicular to the axis of the branch at the time of perching, the perching maneuver is succeeded. This demonstrates that our approach is robust to changes in the related angle between the direction of the flight and the axis of the branch. The system is able to compensate until 10° of deviation (see Fig. 5-F, in the revised paper). Due to the difficulties in controlling large flapping wing robots, it is impossible to reach the branch completely perpendicular, however our claw-leg mechanism is able to deal with these inaccuracies.
- The little variations around the reference value in the pitch angle θ at the time of perching demonstrate that the regulation of this state at the time of perching also works well. This state is very important in order to slow down the flapping wing bird and change the posture just before the perching maneuver.
- The variations in the position Z demonstrates that we can reach autonomously a target height with high precision. Even so, it exists an error position that demonstrates the inaccuracies in the control of altitude due to the inherent oscillatory behavior of flapping wing robots (see Figure 5.E). This fact motivates the use of our approach, an active leg control with a branch detection sensor able to compensate inaccuracies of at least $\pm 10cm$. Moreover, it verifies the importance of the proposed approach which is, for the first time, implemented on a flapping-wing robot for perching maneuvers with claw.
- The variations in the position Y are within a maximum value of 0.15 m. This is important in order to know the precision in the perching maneuver. Also, it demonstrates that the control loop related to Y could regulate the robot with error way less than the length of the branch.

In addition, we have analyzed the results of the failed flights, see Table 7, (in reply-to-comments letter, below). We notice that in Fail (I) and Fail (II), the variables V_x , Yaw angle, Pitch angle and $Z(m)$ are in the range of the successful perching, however, the variable Y is very high, so the lateral control is not accurate enough and the ornithopter was not able to reach the desired branch. Figure 3 (in reply-to-comments letter, below) shows the deviation in the lateral movement in comparison with the 4 experiments that are successful using the rigid branch. On the contrary, in the experiment Fail (III), all the variables, except for Z , are in the admissible range of values of the successful perching experiments. However, the system was not able to grasp the branch. The reason was that the variable Z was out of the range of the successful perching flights. Then, the system reached the X and Y position of the branch with a very high altitude. In this experiment the claw grazed the branch but does not grasp it.

Table 6: Statistical analysis of flight results.

	V_x [m/s]	Yaw angle [°]	Pitch angle [°]	Y [m]	Z [m]
Soft branch (I)	2.43	-8.3	23.6	-0.05	2.06
Soft branch (II)	2.07	-4.6	25.1	-0.23	2
Branch (I)	2.52	1.9	26	-0.02	1.96
Branch (II)	2.54	3.1	31.8	-0.07	1.95
Branch (III)	2.34	4	25.8	-0.15	2.04
2nd Robot	2.8	0.52	30.3	0.02	1.99
Mean	2.45	-0.56	27.1	-0.08	2.0
Standard deviation	0.24	4.85	3.21	0.09	0.04
Reference value	2.5	0	30	0	2
Worst value	2.07	-8.3	23.6	-0.23	2.06

The ornithopter subsequently fell into the net after the branch. The control of altitude was not accurate enough and the actuated leg was not able to compensate the misalignment between the center of the branch and the claw. Figure 4 (in reply-to-comments letter, below) shows the altitude movement of different flights showing that in the experiment Failed (III), the variable Z is the highest of all at the position of perching.

These experiments show the importance of having a very accurate control of all the variables of the system and the difficulty of succeeding in the perching of flapping wing robots.

Table 7: Analysis of failed flight (added to Table S5)

	V_x [m/s]	Yaw angle [°]	Pitch angle [°]	Y [m]	Z [m]
Fail (I)	2.38	0.473	28.22	-0.523	1.992
Fail (II)	2.22	1.67	26.89	-0.518	2.06
Fail (III)	2.45	-0.1	28.23	-0.231	2.09

Figure 3: Analysis of failed flight.

Figure 4: Analysis of failed flight.

The analysis and figures have been added in the supplementary material as Table S4 and Fig. S18. A mention in the main text has also been made:

“We report that branch perching was achieved in 6 out of 9 perching flight tests, and that no damage occurred in the successful flights. The 3 failed perching maneuvers narrowly miss the branch, and are reported in Table S4 and Fig. S18.”

We have re-edited videos S6 and S7 to show the 2 soft perches and the 3 perches on the hard branch. We have also extended the videos to show the robots until rest (except for hard branch 1 where the raw footage was not recorded any longer).

Comment 2 *The authors state in the introduction: “We also introduce a ***development process***, consisting of four experimental steps which lead to autonomous perching flights.” The stars indicate bold text. This sounds as if this development process is a major contribution of the article. Looking at the development process, illustrated in Fig. 4, it is very sensible, but it does not seem so innovative that it merits mentioning as a major contribution. I suggest that the authors slightly downplay this particular contribution in the introduction.*

We thank the reviewer for this comment. Looking back at our research, we fully agree that this contribution was exaggerated and have reduces the emphasis on this point as follow:

“~~We also introduce a development process, consisting of four experimental steps which lead to autonomous perching flights.~~

We also present the development process, consisting of four experimental steps, which leads to autonomous perching flights.”

Comment 3 *In the discussion, the authors evaluate the use of the re-opening for taking off again: “Then, the leg actuation should be improved to execute a jump forward, synchronized with claw re-opening. Secondly, given sufficient ground clearance, the vehicles could fall from the branch and perform a controlled flight recovery.” However, in the same time, opening takes around 20 seconds. . . How can a jump be synchronized with such a slow opening?*

We agree with the reviewer that synchronizing the slow opening of the claw with the jump of the system is complex and it has to be studied in depth. The main contribution of the work is the integration of all the subsystems and perching with a flapping-wing robot, including a claw-opening mechanism. This mechanism is featured for 2 reasons (as mentioned in the Claw Opening section):

1. Initially open the claw before perching
2. Re-open once perched for future take-off maneuvers.

As it is said in the paper, this mechanism is not needed in the perching maneuver itself, and taking-off from the branch will be the focus of our future works.

In previous works, we have explored the possibility to change the desired posture of the system once the system is perched (see references (31), (39)). We have in mind to use these works in the taking-off approach. The idea is to change the position of the system in order to obtain a stable configuration where, with almost zero friction, the ornithopter remains perched in the branch. Once this posture is achieved, the claw starts to open slowly, decreasing the friction between the claw and the branch. In the very last second, where the claw is almost opened and the friction is very low, we propose two approaches:

1. Move the system in order to get out of its stable position, fall and perform a controlled flight recovery (only if there is sufficient distance between the branch and the ground)
2. Execute a jump forward and start flapping the wings at maximum speed to recover the flight. For this second approach, the claw-leg mechanism has to be changed in many aspects while for the first one, the claw can be used as it is

We have proposed these two approaches to be studied in the future, and in this way, motivate the need of having an opening mechanism in the proposed claw. The main text has been clarified to include the answer to this question as follows:

“Lastly, take-off from the branch will be the focus of future work. While the leg-claw system can fully re-open from the branch, the absence of initial velocity currently impedes taking-off flight. Several strategies are envisioned. First, achieving the desired correct the posture once the system is perched to facilitate obtain a stable configuration, where with almost zero friction, the claw can re-open slowly. This would leverage the previous work addressing balancing challenges in (31, 39). Then, the leg actuation should be improved to execute a jump forward, synchronized with claw re-opening, and with maximum flapping thrust. Secondly, given sufficient ground clearance, the vehicles could fall from the branch and perform a controlled flight recovery. ”

Comment 4 *My compliments for Fig 1. It is beautiful. I do think it would be even more illustrative to indicate the size and weight of the flapping wing robot in some way (perhaps simply with text for wingspan and total mass), add an motion tracking camera to the image, and adapt “last-meter sensor” to something like “linear vision camera for last-meter corrections”.*

Thank you for the kind comment. This is an excellent suggestion, the figure has been updated with indication to the size, weight and chord length. The mention to the last-meter sensor has been changed. We have also added a motion capture camera pictogram and mention. Please bear in mind that the full scene is presented in fig S8.

Please see Fig. 5 below.

Figure 5: Intro figure

2.2 Minor comments

Comment 1 “design can lock” to “design that can lock”

Thank you, we have updated the main text.

Comment 2 Ref 6 seems to miss a title and venue.

It was a mistake. It is corrected in the revised version:

“Zufferey, R., Siddall, R., Armanini, S. F., and Kovac, M. (2022). *Between Sea and Sky: Aerial Aquatic Locomotion in Miniature Robots*. Springer, Vol. 29. <https://doi.org/10.1007/978-3-030-89575-4>”

Comment 3 “The unsteady aerodynamics of the vehicles lead to hard-to-model dynamics and therefore inaccurate positioning” - Is the flapping in normal flight conditions not quite stereotypical? Or do you see large variations in the flapping amplitude in nominal flight conditions? I think it can be said that the flapping motion leads to additional disturbances and less accurate control.

The instantaneous lift and drag forces of the wings are hard to model (aero-elastic effects, body drag, membrane wing). This indeed results in less accurate control and therefore increases the difficulty to position the robot accurately. We have corrected the main text to closer match the reviewer's suggestion:

~~"The unsteady aerodynamics of the vehicles lead to hard-to-model dynamics and therefore inaccurate positioning. The hard-to-model, unsteady aerodynamics of the flapping wing motion lead to less accurate control and therefore less accurate positioning."~~

Comment 4 *"The large size of ornithopters further increases the perching difficulty." - The definition of ornithopter is an aircraft that flies by flapping its wings (<https://en.wikipedia.org/wiki/Ornithopter>). This does not have any consequence for its size (ornithopters can be as small as Harvard's robobee). Please be more specific here, e.g.: "Moreover, perching is more difficult for large (> 1m wing span) ornithopters." Similarly on page 10: "While ornithopters are passively stable, it" - tailed ornithopters are typically passively stable, tailless ones are not.*

This is an excellent note. Ornithopters stem from orni, or bird, explaining why this term is used mostly for bird-scale robots in the literature. But according to the wikipedia definition, scale is indeed not part of the definition. Any ambiguity should be removed and we follow the suggestion to be more exact as follows:

~~"The large size of ornithopters further increases the perching difficulty. Moreover, perching is more difficult for large (>1m wing span) ornithopters."~~

~~"While the tailed ornithopters in this research are passively stable,"~~

Comment 5 *The authors state on smaller flapping wing robots: "(8, 9), however they suffer from limited payload." Later, they state on larger flapping wing robots: "Flapping-wing robots also face additional constraints such as stringent payload restrictions and oscillations that need to be addressed." This forms a bit of a paradox. . . Is the flapping-wing robot size selected by the authors the "sweet spot", where payload restrictions are interesting and not limiting? Or can the current work lead to insights that will help also smaller flapping wing robots to perch? As I interpret it, the authors work with a larger system, since it is currently easier to add the necessary hardware for perching. A more detailed / subtle reflection would be appreciated.*

This is indeed a good interpretation of the choice of size. At the small-scale, the integration of an additional mechanical system is extremely challenging. For example, only very recently has some level of additional mechanical functionality been added to smaller flapping-wing robots [R6] (in reply-to-comments letter, below); and only without extra actuation/energy storage. Flapping flight (and the perching task) scales unfavorably as explained at the end of the 2nd paragraph of the intro. Therefore it makes sense to limit how large the ornithopters are. Additionally, in the context of flight in a controlled environment, any robot larger than selected here becomes unpractical or impossible to fly in a motion capture system. This, for experimental reasons, is the stricter limit in our case.

In summary, the scale choice represents a good balance between ease of manufacturing and flight space but would transfer well to smaller sizes. Regarding adaptability to smaller systems, it would require additional manufacturing/design efforts but would remain possible as long as servo-actuators are commercially available at that size. From our experience, we hypothesize that applying this method to a robot down to 200 g total mass should be feasible.

In light of the above, we have updated the main text to clarify the two quotes from the reviewer. We also add a descriptive sentence regarding the trade-off.

“Small-scale birds and robots that are capable of hovering circumvent this issue (8, 9), however they suffer from limited payload and increased manufacturing complexity.”

“Compared to fixed-wing robots, flapping-wing robots face additional constraints such as stringent more payload restrictions and oscillations that need to be addressed.”

“This class of robots is large, with a 150 cm wingspan and a 500 g empty weight. This size represents a good trade-off between manufacturing and testing and scaling constraints, i.e. the robots utilizes commercial components, widely-employed prototyping methods yet is small enough to fly within a motion capture lab space. These metrics introduce a basis for the sizing of the specific solutions.”

[R6] Crawl and Fly: A Bio-Inspired Robot Utilizing Unified Actuation for Hybrid Aerial-Terrestrial Locomotion, DOI:10.1109/LRA.2021.3099246

Comment 6 *How do the perching multirotors compare to the current platform in terms of size and mass?*

A list of multiple rotors that have been demonstrated to perch is given in table 8. Robots with only 2 rotors (helicopter) to 6 rotors (hexacopter) are presented. The weight varies from light 200 g systems to over 2 kg. The absence of wings makes multirotors relatively compact with even the biggest vehicle not exceeding 550 mm in vehicle dimension.

We have added the missing references in the main text, as part of the discussion of different gripper mechanisms, and reordered the paragraph:

“Attachment to a surface is an important issue, addressed differently with systems such as micro-spines (14, 15), spines (16), fiber-based adhesive (17), or nature-inspired mechanical grippers (18, 19). Multirotor UAVs are able to hang passively from branches of various diameters (20), or sit passively on branches (21). Attachment under beams was also shown with quadcopters, based on a bistable clamping mechanism (22). Researchers have also investigated how to perch robots to point locations. For example, multirotor UAVs, capable of hovering, were employed to perform localized perching with vision-based feature identification (23, 24).”

New references has been updated in the reference section of the revised paper.

Table 8: Perching multirotor weight and size

Robot	Year	Method	Weight	Size
1 HEDGEHOG	2021	Spines	400 g	~ 200 mm
2 SNAG	2021	Claws	750 g	~ 170 mm
3 Fin Ray Grasper	2022	Claws	295 g	290 mm
4 DJI F450 UAV	2019	Landing gear	1400 g	450 mm
5 DJI FlameWheel	2016	Claws	2300 g	550 mm
6 Microspine Grapple	2022	Micro-spines	1766 g	450 mm
7 Sarrus helicopter	2016	Claws	200 g	~ 150 mm

Comment 7 *“The geometry of the claw is modeled to offer maximum friction at the branch level while having a large opening angle and low trigger force.” - Is this probably the maximum level of friction?*

The aim of the claw model is to find a design that :

- Maximizes clamping force. For that, the spring's origin needs to be in a specific location and angle to maximize its force yet not plastically deform
- Offer a large opening angle. Forces could be higher if the claw did not travel as much, but the claw still needs to open wide to capture the branch.
- Minimize trigger forces. The 4 degree negative angle determines how much force is required to trigger. Ideally this value would be close to zero, yet we have to take into account flight vibration that might inadvertently close the claw and manufacturing precision. 4 degrees was found to give a reasonable compromise.

"The geometry of the claw is modeled to offer maximum friction clamping force at the branch level while also having a large wide opening angle to capture the branch and low trigger force sufficiently low trigger force to ensure reliable closing on impact. "

Comment 8 *"The claw dimension is smaller than the flight position accuracy" What do you mean here? How was the flight position accuracy measured / determined?*

The flight position accuracy was measured by comparing the commanded desirable height with the real value which was measured by the motion capture system (opti-track system). This was already estimated from previous works (29) (reference in the revised paper) to be around $\pm 10-15$ cm. The oscillations clearly deteriorate the accuracy significantly. Similar results is observed again in this research as stated in the flight experiments section.

Here, we wanted to explain that without actuation of the leg for adding extra workspace to the claw, the grasping is unlikely to be successful. We rewrote the sentence to quantify better:

"The claw dimension is smaller than the flight position accuracy The mechanical contact zone of the claw is 5 cm, which is lower than flight altitude accuracy, previously estimated to $\pm 10-15$ cm (29). As this is insufficient to reliably touch perch on the target "

Comment 9 *Page 8: "Future outdoor perching maneuvers in cluttered environments will benefit from improved sensing methods, e.g. full 2D or 3D vision systems (22)." - The authors could refer here to work with event or CMOS cameras on flapping wings here as well:*

- Eguíluz, A. G., Rodríguez-Gómez, J. P., Paneque, J. L., Grau, P., de Dios, J. M., & Ollero, A. (2019, November). Towards flapping wing robot visual perception: Opportunities and challenges. In 2019 Workshop on Research, Education and Development of Unmanned Aerial Systems (RED UAS) (pp. 335-343). IEEE.*
- De Wagter, C., Tijmons, S., Remes, B. D., & de Croon, G. C. (2014, May). Autonomous flight of a 20-gram flapping wing mav with a 4-gram onboard stereo vision system. In 2014 IEEE International Conference on Robotics and Automation (ICRA) (pp. 4982-4987). IEEE.*

Thank you for the suggestion - the references have been added:

"Future outdoor perching maneuvers in cluttered environments will benefit from improved sensing methods, e.g. 3D vision systems (25), stereo CMOS-based or event-based cameras (33, 34). "

Comment 10 *“Selection of a suitable trajectory in which no control limitations appear, such as saturation or limited stability margins” - What trajectories are / are not possible and why?*

Thank you for bringing up this point. This comment is stated inside “Robot avionics and control” section to show the design steps for performing experiments. The flapping robot has a specific flight envelope (the airspeed and load factor or atmospheric density, often simplified to altitude or conditions that lead to a successful flight including the percentage of flapping frequency, payload, etc.). More information is given in Fig. 7 of the following reference:

Zufferey, Raphael, Jesús Tormo-Barbero, M. Mar Guzmán, Fco Javier Maldonado, Ernesto Sanchez-Laulhe, Pedro Grau, Martín Pérez, José Ángel Acosta, and Anibal Ollero. “Design of the high-payload flapping wing robot e-flap.” IEEE Robotics and Automation Letters 6, no. 2 (2021): 3097-3104.

where one can see what trajectory is attainable with varying loads. These conditions imply that the workspace of the flight is limited and not all the initial conditions nor all flight trajectories are accessible. A counter-example is as follows: Putting the bird with the horizontal pose on the launcher and target 3 meters Z, height, set-point regulation. The robot with that initial pose cannot pave a trajectory to increase 3 meters height in 14 meters longitudinal distance. If we set the mentioned condition, from the beginning of the flight, the tail and flapping frequency go to saturation condition and therefore may not succeed, because the trajectory is outside the feasible flight envelope.

The following text has been added after Fig. 4, highlighted in yellow, to clarify this point.

“Selection of a suitable trajectory in which no control limitations appear, such as saturation or limited stability margins. Aggressive maneuvers or increases in height of more than 3 meters in the 14-meter longitudinal test-bed workspace are unsuitable trajectories which would force the flapping frequency and tail into saturation and miss the target. Therefore, the arrangement of the initial condition of the robot on the launcher, launching speed, angle, and desired Z height position should be reasonably chosen.”

Comment 11 *Page 10: On one hand / on the other hand: Are these really opposing things?*

Thank you for the suggestion; changed in the main text:

“~~On the one hand~~Indeed, flapping is needed to maintain a stable flight resulting in oscillations in the controlled height. ~~On the other hand,~~As in most fixed-wing aircraft, the altitude dynamics are non-minimum phase, which inherently limits the achievable settling time. ”

Comment 12 *Page 11: “up to 5m/s” Where can we see that? Fig 5E goes up to 4 m/s.*

Thanks for spotting this mistake, the main text has been corrected to 4 m/s.

Comment 13 *“results show that the system can tolerate a branch deviation in a range of $\pm 10-20^\circ$ depending on the leg pitch orientation” - Is this not about yaw?*

Yes, this could be interpreted that the claw tolerates $\pm 10-20^\circ$ deviation in yaw angle since the branch is fixed and the bird is supposed to perch preferably perpendicular. The following text has been added after that sentence to clarify this point based on the reviewer’s suggestion:

“The results show that the system can tolerate a branch deviation about yaw in a range of $\pm 10-20^\circ$, which also depends on the leg pitch orientation, where 90° is a horizontal leg. Therefore, in case of a small yaw

angle deviation from the perpendicular direction, the claw can still adequately hold onto the branch, at 4m/s perching speed."

REVIEWERS' COMMENTS

Reviewer #1 (Remarks to the Author):

I appreciate the author's detailed response to my previous comments. In my view, this revised manuscript has been substantially improved. I recommend the paper to be published in the present form.

Reviewer #2 (Remarks to the Author):

The authors have done a wonderful job addressing all comments and consequently modifying the article. I really appreciate the added analysis of failed maneuvers. I have no further remarks.

Nature Communications 2022 - Response to reviewers

Manuscript **NCOMMS-22-28797**

How ornithopters can perch autonomously on a branch

November 2022

We would like to sincerely thank the reviewers for their evaluation of our work. We are happy to read that they found that our previous revision satisfactorily answered their comments.